# Drivers of Multicentury Trends in the Atmospheric $CO_2$ Mean Annual Cycle in a Prognostic ESM

Jessica Liptak[1], Gretchen Keppel-Aleks[1], and Keith Lindsay[2]

[1]Department of Climate and Space Sciences and Engineering, University of Michigan, Ann Arbor, MI, USA
[2]Climate and Global Dynamics, National Center for Atmospheric Research, Boulder, CO, USA

*Correspondence to:* Jessica Liptak (liptak@umich.edu)

**Abstract.** The amplitude of the mean annual cycle of atmospheric $CO_2$ is a diagnostic of seasonal surface-atmosphere carbon exchange. Atmospheric observations show that this quantity has increased over most of the Northern Hemisphere (NH) extratropics during the last three decades, likely from a combination of enhanced atmospheric $CO_2$, climate change, and anthropogenic land use change. Accurate climate prediction requires accounting for long-term interactions between the environment and carbon cycling, so analysis of the evolution of the mean annual cycle in a fully prognostic Earth system model may provide insight into the model sensitivity to the multi-decadal influence of environmental change on the carbon cycle.

We analyzed the evolution of the mean annual cycle in atmospheric $CO_2$ simulated by the Community Earth System Model (CESM) from 1950 to 2300 under three scenarios designed to separate the effects of climate change, atmospheric $CO_2$ fertilization, and land use change. The NH $CO_2$ seasonal amplitude increase in the CESM mainly reflected enhanced primary productivity during the growing season due to climate change and the combined effects of $CO_2$ fertilization and nitrogen deposition over the mid- and high latitudes. However, the simulations revealed shifts in key climate drivers of the atmospheric $CO_2$ seasonality that were not apparent before 2100. $CO_2$ fertilization and nitrogen deposition in boreal and temperate ecosystems were the largest contributors to mean annual cycle amplification over the midlatitudes for the duration of the simulation (1950–2300). Climate change from boreal ecosystems was the main driver Arctic $CO_2$ annual cycle amplification between 1950 and 2100, but $CO_2$ fertilization had a stronger effect on the Arctic $CO_2$ annual cycle amplitude during 2100–2300. Prior to 2100, the NH $CO_2$ annual cycle amplitude increased in conjunction with an increase in the NH land carbon sink. However, these trends decoupled after 2100, underscoring that an increasing atmospheric $CO_2$ annual cycle amplitude does not necessarily imply a strengthened terrestrial carbon sink.

## 1 Introduction

The amplitude of the mean annual cycle of atmospheric $CO_2$, an indicator of the seasonal cycle of terrestrial and ocean carbon exchange, has increased over the Northern Hemisphere (NH) since observational records began in the late 1950s (Pearman and Hyson, 1981; Cleveland et al., 1983; Bacastow et al., 1985; Conway et al., 1994; Keeling et al., 1996; Randerson et al., 1997; Graven et al., 2013; Liu et al., 2015). The largest increases of 40–50% have been observed over the northern high latitudes via surface monitoring (Keeling et al., 1996) and from aircraft observations of the free troposphere (Graven et al., 2013).

The amplification of the atmospheric $CO_2$ annual cycle primarily reflects enhanced net exchange of $CO_2$ with land surfaces rather than with the ocean (Manning, 1993). Land-atmosphere $CO_2$ exchange is highly seasonal, especially in the NH mid- and high latitudes where photosynthesis draws down $CO_2$ in the spring and summer, and net ecosystem respiration returns $CO_2$ to the atmosphere (e.g., Komhyr et al., 1985; Enting and Mansbridge, 1989; Nemry et al., 1996; Dettinger and Ghil, 1998).

Because atmospheric $CO_2$ observations are characterized by high precision and accuracy, the gradual, multi-decadal increase in the seasonal amplitude provides a unique observational target for Earth system models (ESMs) intended to predict the long term carbon cycle-climate evolution. ESMs enable the study of long-term effects of natural and anthropogenic forcing on the terrestrial carbon cycle, since they include mechanistic representations of the carbon cycle by coupling land surface models that explicitly resolve biogeochemical processes with models of the atmosphere, ocean, and other components of

the climate system (Claussen et al., 2002). An advantage of using a coupled model is that feedbacks between the physical climate and biogeochemistry are represented in a self-consistent framework. This is crucial since carbon fluxes are inherently linked to the physical climate; for example, a change in gross primary productivity (GPP) will be associated with changes in evapotranspiration, which feeds back on metrics such as humidity, cloud cover, and precipitation. Moreover, in a fully-prognostic model, both climate and carbon cycle diagnostics are free to evolve rather than being tied to input data sets that

reflect the contemporary climate. The mechanisms embedded in ESMs to predict future carbon-climate interactions have been identified as the likely drivers of the observed mean annual cycle amplitude increase as described below.

The magnitude of the amplitude increase suggests a dominant role for enhanced primary productivity during the growing season in addition to increased $CO_2$ release during the dormant season (Graven et al., 2013). Greater atmospheric $CO_2$ may facilitate plant carbon uptake through increased water use efficiency (Keenan et al., 2013), and results from Kohlmaier et al.

(1989) and McGuire et al. (2001) suggest that $CO_2$ fertilization adds at least 10% to the $CO_2$ mean annual cycle amplitude trend. Human activity has not only increased the atmospheric concentration of atmospheric $CO_2$, but also modified reactive nitrogen deposition (N-deposition) in ecosystems. Likewise, deposition of nitrogen oxides ($NO_x$) and ammonia from combustion, livestock, agriculture, and industrial sources may augment the supply of soil nitrogen available for fixation by plants (Prentice et al., 2001), alleviating a limitation on terrestrial GPP (Lloyd, 1999; Norby et al., 2010).

Climate change-induced warming and lengthening of the growing season may also stimulate GPP and increase the seasonality of net exchange. Keeling et al. (1996) proposed that increased terrestrial $CO_2$ uptake from a longer high latitude growing season has driven the amplification of the $CO_2$ annual cycle, since the trends in the $CO_2$ annual cycle amplitude strengthen moving northward, and the greatest warming has occurred during the winter and spring over the northern high latitudes. Findings by Randerson et al. (1999), McDonald et al. (2004) and Barichivich et al. (2013) support the hypothesis that longer

growing seasons enhance spring $CO_2$ uptake and annual cycle amplitudes over the Arctic. This effect may be counteracted by the fact that early growing season onset may lead to growing season moisture deficits that reduce terrestrial productivity later in the growing season (Angert et al., 2005; Buermann et al., 2013; Parida and Buermann, 2014). Model evidence suggests that climate-driven shifts in vegetation cover can also enhance GPP. Forkel et al. (2016) showed that the interaction of vegetation dynamics and climate change lead to GPP over NH boreal and Arctic regions that, in turn, drove the observed increases in NH

high latitude seasonal $CO_2$ amplitudes.

In short, there are several potential drivers of the $CO_2$ annual cycle amplification that feed back onto the climate system. Despite the representation of these mechanisms by ESMs, Graven et al. (2013) showed that none of the CMIP5 carbon cycle models was able to simulate the magnitude of the observed increase in atmospheric $CO_2$ seasonality. Since understanding the drivers of the $CO_2$ seasonality is crucial for model development, we used the Community Earth System Model (CESM) to study the contribution of natural drivers of variability in $CO_2$ fluxes to the increasing amplitude by separating the effects of $CO_2$ radiative forcing (climate change), $CO_2$ fertilization and N-deposition, and land use change on the atmospheric $CO_2$ annual cycle amplitude over the NH subtropics, NH midlatitudes, and the Arctic before and after 2100 in an extension of the high-emissions RCP8.5 scenario (Meinshausen et al., 2011).

In addition to revealing potential effects of continued increases in $CO_2$ emissions, anthropogenic nitrogen, and land use change up to 2100, extending the RCP8.5 scenario to 2300 allowed us to assess the behavior of the mean annual $CO_2$ cycle in a warmer climate following stabilization of atmospheric $CO_2$ mole fraction, and a shift in the terrestrial biosphere from a $CO_2$ sink to a source as shown by Randerson et al. (2015).

The questions guiding our analysis of CESM extended concentration pathway simulations are as follows:

1. Does the relative importance of drivers of the $CO_2$ amplitude trend change after 2100? For example, do we see evidence of saturation of the $CO_2$ fertilization effect or evidence of a climatic tipping point after which the $CO_2$ amplitude declines?

2. Do the regional contributions to $CO_2$ mean annual cycle trends change in response to large changes in climate?

3. Does the $CO_2$ annual cycle amplitude scale with the hemispheric carbon sink from NEP as climate and atmospheric conditions evolve in the future?

The CESM provides a unique platform for exploring these questions in that it is one of the few prognostic ESMs to include coupled carbon-nitrogen biogeochemistry and diagnostic atmospheric $CO_2$ variability. This paper is organized as follows: First, we discuss the ability of the CESM to capture observed changes in the mean $CO_2$ annual cycle amplitude throughout the NH. Second, we describe how climate change, $CO_2$ fertilization and N-deposition, and land use change impact the NH $CO_2$ annual cycle amplitude in the CESM before and after 2100. Third, we examine how forcing from different regions contributes to the amplitude changes attributed to each driver. Finally, we discuss our results and provide recommendations for future analysis.

## 2   Methods

### 2.1   Model

We analyzed simulations from the Community Earth System model with coupled biogeochemistry (CESM1(BGC); Hurrell et al. (2013)) to explore the role of environmental change on land-atmosphere carbon exchange. The Community Atmosphere Model (CAM, version 4; Neale et al. (2013)) and the Community Land Model (CLM, version 4; Lawrence et al. (2012)) were the most

important components for our research, but all components of the model, including physical and biogeochemical ocean processes and sea ice processes, were interactive in the model configuration. The CAM4 was run on a $0.94° \times 1.25°$ finite volume grid with 26 vertical levels. The model simulated climate conditions, including temperature, precipitation, and humidity that provide important boundary conditions for land biogeochemistry. Moreover, the CAM4 directly simulates three-dimensional transport of atmospheric $CO_2$, as well as separate $CO_2$ tracers derived from fossil fuel emissions, land exchange, and ocean exchange.

The CLM4 exchanged fluxes of sensible and latent heat, momentum, moisture, radiation, and terrestrial carbon with the CAM4, and was run at the same horizontal resolution. Biogeochemistry is represented in CLM4 by a prognostic carbon-nitrogen model (CLM4CN, Thornton et al. (2007)) and fire model adapted from the Thonicke et al. (2001) model. We note that important high latitude processes, such as permafrost carbon dynamics, were not simulated in the CLM4, meaning that the model may have underestimated both the seasonal dynamics of soil carbon fluxes and the long-term dynamics of permafrost melt and the subsequent radiative feedback onto the climate system (Koven et al., 2015). In our analysis, we used the CLM4 net ecosystem productivity (NEP), defined as the difference between GPP and total respiration (autotrophic + heterotrophic) to calculate the atmospheric $CO_2$ annual cycle amplitudes described in Section 2.3.

## 2.2 Experiments

Three CESM simulations were run from 1850 to 2300 to separate the effects of climate change, $CO_2$ fertilization and N-deposition, and land use change. The mole fraction of $CO_2$ in the atmosphere is prescribed according to the RCP8.5 and ECP8.5 scenario described by (Meinshausen et al., 2011), and it is this value that controls radiative forcing as well as $CO_2$ fertilization. However, the CESM retains a separate, spatially-varying $CO_2$ tracer that is a diagnostic passive tracer of land, ocean, and fossil fuel carbon fluxes; the additional carbon exported from the land and ocean to the atmosphere does not exert any radiative forcing on the climate.

The degree of coupling between $CO_2$ biogeochemistry and radiative forcing differed across the three runs. In the first simulation, denoted FullyCoupled, the imposed $CO_2$ was radiatively active, and additional anthropogenic radiative forcing resulted from prescribed $CH_4$, chlorofluorocarbons, ozone, and aerosols. In this simulation, the increasing $CO_2$ was also biogeochemically active, meaning it contributed to $CO_2$ fertilization. Transient land use change (LUC) from agriculture and wood harvest, and land and ocean Lamarque et al. (2010)) were applied through 2100, then held at 2100 values thru 2300. Keppel-Aleks et al. (2013) and Lindsay et al. (2014) provide additional descriptions of the model configuration and analyses of the FullyCoupled simulation during the 20th century. In the second simulation (NoRad), radiative forcing from $CO_2$ and other species was fixed at 1850 values, but the changing $CO_2$ mole fraction interacted with biogeochemistry via $CO_2$ fertilization. LUC and N-deposition were likewise prescribed as in FullyCoupled. Randerson et al. (2015) also details the design of the FullyCoupled and NoRad (referred to as "NoCO$_2$Forcing") simulations through 2300. We isolated the impact of climate change on the mean annual $CO_2$ cycle by taking the difference between the FullyCoupled and NoRad simulations. The third simulation (NoLUC), was configured identically to FullyCoupled with the exception that LUC was held constant at 1850

values; therefore, LUC effects on terrestrial carbon exchange were determined from the difference between FullyCoupled and NoLUC.

Variations in fractional coverage, albedo, nutrient limitations, and surface energy fluxes among trees, grasses, and crops may enhance or oppose the effects of climate change and $CO_2$ fertilization on the atmospheric $CO_2$ mean annual cycle amplitude. These plant functional type (PFT)-based changes were approximated by prescribing transient land cover change through 2100 in FullyCoupled and NoRad based on annual fractional transition among primary vegetation, secondary vegetation, pasture (grazing land), and crops described by Hurtt et al. (2006) with CESM PFTs detailed in Lawrence et al. (2011). The crop model was inactive in the CESM simulations, and the crop PFT in Hurtt et al. (2006) data was specified as unmanaged grass (Lindsay et al., 2014). Therefore, our CESM results do not include anthropogenic influences on $CO_2$ seasonality or agricultural intensification. Moreover, these simulations were run without dynamic vegetation, which potentially damps feedbacks that could contribute to changes in the $CO_2$ annual cycle through 2300.

## 2.3  Mapping atmospheric $CO_2$ from surface fluxes

Although the CESM simulated the three-dimensional structure of atmospheric $CO_2$, we used a pulse-response transport operator to separate imprints of $CO_2$ fluxes from different regions on the atmospheric $CO_2$ variations. The transport operator was developed using the GEOS-Chem transport model (version 9.1.2, Nassar et al. (2010)). GEOS-Chem was configured as in Keppel-Aleks et al. (2014) on a $4° \times 5°$ horizontal grid with 47 vertical layers, and forced with meteorology fields from the 3–6-hourly Modern Era Retrospective-Analysis for Research and Applications (MERRA) reanalysis dataset (Rienecker et al., 2011). A tagged 1 Pg C $\text{month}^{-1}$ pulse was released for each of the 20 terrestrial source regions in Fig. 1 for each calendar month, and allowed to decay for 60 subsequent months. Each 1 Pg C $\text{month}^{-1}$ pulse was distributed spatially according to monthly fluxes from the Carnegie-Ames-Stanford Approach (CASA) fluxes from Olsen and Randerson (2004).

At a given location, the magnitude and phasing of the atmospheric $CO_2$ response of the pulse depends on the characteristics of atmospheric transport (Fig. 2). For example, at Barrow (BRW) in Northern Alaska, a 1 Pg pulse released in Boreal North America (NBNA) in the winter months (December–January) has a large impact on atmospheric $CO_2$ during the first 1–2 months after a pulse is released (2 ppm, Fig. 2a), but more vigorous vertical mixing in the summer months reduces the imprint to 0.5 ppm. In contrast, when the pulse is released from temperate North America (ETNA, WTNA), there is a phase lag of 2–3 months (Fig. 2b,c), and when the pulse is released from the Amazon (AMZN), there is a delay in the peak response at BRW of at least 4 months (Fig. 2d). Following the 12-month period in which pulses were released, the signals were allowed to decay for 60 subsequent months, at which point $CO_2$ was well-mixed in the atmosphere (Fig. 2a–d). We then sampled GEOS-Chem at the locations of 41 NOAA cooperative $CO_2$ flask sample sites (Dlugokencky et al. (2013); Table 1, Fig. 1) for each of 72 total months simulated. This resulted in a $CO_2$ transport operator matrix with dimensions $N_{\text{reg.}} \times N_{\text{obs.}} \times N_{\text{mon.}}$.

We used monthly mean NEP from the CESM to derive atmospheric $CO_2$ from the pulse response function. We aggregated NEP fluxes from CLM4 to the spatial scale of the 20 source regions (Fig. 1), and used matrix multiplication to propagate these fluxes to atmospheric $CO_2$. We calculated the monthly mean $CO_2$ mole fraction at the observation sites (e.g., blue lines in Fig. 3) by summing over the instantaneous contributions from all regions, and the background contributions from fluxes

released during the 60 previous months, to get a $CO_2$ response matrix with dimensions ($N_{obs.} \times N_{mon.}$). We analyzed both the $CO_2$ fields from global fluxes and $CO_2$ patterns influenced only by larger regions representing Arctic, boreal, temperate, subtropical, tropical, and Southern Hemisphere (SH) ecosystems. We calculated the $CO_2$ annual cycle amplitude values as the peak-to-trough differences in $CO_2$ summed over each component region (e.g., the $CO_2$ annual cycle amplitude at a given station from pulses emitted from the Arctic was calculated as the peak-to-trough difference in the sum of $CO_2$ from pulses emitted by the blue regions in Fig. 1). We note that our analysis focuses on surface observations of atmospheric $CO_2$, and does not include aircraft measurements.

The advantage of the pulse-response method is that we can efficiently compute the regional contribution to changes in atmospheric $CO_2$; it would be prohibitively expensive to run a full atmospheric transport model for each of the regions separately for 350 years. However, using this simplified transport operator introduces errors. To evaluate the pulse-response method, we show a comparison in which we have generated $CO_2$ using net ecosystem exchange (NEE), which includes fire, harvest, and land use fluxes (Fig. 4), since the land $CO_2$ tracer in the CAM4 is derived from NEE (despite that we use NEP for subsequent analyses). The errors are generally less than 2 ppm between the full transport and pulse response calculations due to different model boundary layer schemes and atmospheric transport (Fig. 4c). We note that the largest differences were during the last century of the simulation, which likely was due to shifts in atmospheric transport in response to the dramatic climate change in the CAM4. The fact that long-term trends in transport are not simulated by the pulse-response approach is one of the major sources of bias. In a site-by-site comparison (Fig. 3), the increasing bias through 2300 appears to be due to amplification of existing biases in the pulse-$CO_2$ compared to the full transport-$CO_2$. A second source of uncertainty is that the spatial distribution of fluxes within each region is different in CLM compared to CASA. We expect that this has a minimal impact based on results from Nevison et al. (2012), who showed that a similar pulse response code using different transport models did a reasonable job ($r^2 = 0.8$) of simulating the fossil fuel influence on $CO_2$ despite that fossil fuel emissions show a vastly different spatial configuration than do ecosystem fluxes. In our analysis, we aggregate the sites into high-, mid-, subtropical, and tropical latitude belts to minimize local effects at individual sites and instead to focus on large-scale trends owing to broad patterns of changing fluxes.

We assessed the validity of the assumption to model only the land contributions to trends in the mean annual cycle of $CO_2$ by calculating the $CO_2$ amplitudes in the CAM land and ocean tracers. We found that the contemporary peak-to-trough amplitude in the ocean tracer averaged across our high latitude stations was 2 ppm (in contrast to 10 ppm in the land tracer). Although both the land and ocean amplitudes grow with time, by 2300, the high latitude ocean tracer had an amplitude of 3 ppm, only 18% of the land amplitude for this time period.

## 2.4 Atmospheric $CO_2$ timeseries analysis

To place the $CO_2$ annual cycle amplitudes simulated by the CESM in the context of observations, we quantified observed and simulated $CO_2$ annual cycle amplitude at NOAA observatories before aggregating amplitudes across four latitude bands spanning 60°–90°N (NH high latitudes), 40°–60°N (NH midlatitudes), 20°–40°N (NH subtropics), and 0°–20°N (NH tropics). We identified a subset of stations in the NOAA Global Monitoring Division (Conway et al., 1994) and Scripps Institute

of Oceanography (Keeling et al., 2005) networks with better than 95% temporal coverage of monthly mean values from 1985–2013 (gray circles in Fig. 1). The trends at these stations were calculated iteratively as a second-order polynomial, as described by Keppel-Aleks et al. (2013). After subtracting the trend from the raw observations, we calculated the peak-to-trough amplitude ($A^{Obs}$) for each calendar year in which observations existed. We then aggregated $A^{obs}$ from all stations within the specified latitude bands to determine a regionally averaged amplitude.

We calculated the regional $CO_2$ amplitudes for the fully coupled simulations ($A^{FC}$) using a nearly identical methodology. However, due to the length of the simulated timeseries, we detrended the data in ten-year increments. For CESM output, we used only the sampling locations with greater than 95% temporal coverage for comparison with the observations (Fig. 1, gray circles), but aggregated amplitudes at a larger set of marine boundary layer observatories when assessing future trends (Fig. 1, black circles). Due to the flexible transport operator, we separately calculated amplitudes from NoRad ($A^{NoRad}$), and NoLUC ($A^{NoLUC}$) simulations, and were further able to simulate only the contribution from specified ecosystem types. The contribution of climate change to the $CO_2$ mean annual cycle amplitude ($A^{Clim}$) was calculated from the difference between $A^{FC}$ and $A^{NoRad}$. Likewise, the LUC contribution to the annual cycle amplitude was calculated from the difference between $A^{FC}$ and $A^{NoLUC}$.

# 3   Results

## 3.1   Trends in observed and modeled $CO_2$ annual cycle amplitudes

Throughout the NH, the CESM simulated both smaller mean annual cycle amplitudes and a smaller trends in amplitude relative to observations (Fig. 6). The CESM underestimated the magnitudes of $A^{Obs}$ by roughly 50% (Fig. 6b, c), and the 16% relative increase in the hemispheric-average amplitude between 1985 and 2013 estimated by the CESM was somewhat lower than the observed increase of 24%. The 1985–2013 mean $A^{Obs}$ averaged over the whole NH was 10.5 ppm, while $A^{FC}$ was 5.8 ppm. At high latitudes, the observed 1985–2013 value was 15.9 ppm, but only 10.8 ppm in the CESM, broadly consistent with Keppel-Aleks et al. (2013) who showed that the CESM1(BGC) underestimated NH seasonality by 25–40%. Although the CESM simulates low mean annual cycle amplitude throughout the NH, we note that many land models have a low bias in their simulated flux seasonality. For example, TRENDY land models show a 40% deficit in the magnitude of the seasonal cycle (Zhao et al., 2016).

Consistent with the observations, the CESM simulated an increasing amplitude trend with latitude over the NH. However, the meridional gradient in the trend was too weak, leading to small absolute increases over the Arctic. Both the modeled and observed trends in the $CO_2$ annual cycle amplitude were calculated from individual sites whose records date to 1985 (gray circles in Fig. 1). The modeled trend in the $CO_2$ annual cycle amplitude over the high latitudes was 0.05 ppm yr$^{-1}$ (0.43% yr$^{-1}$) for the 1985–2013 period, while the observed trend was 0.09 ppm yr$^{-1}$ (0.57% yr$^{-1}$). Midlatitude and subtropical trends simulated by the CESM were around 0.03 ppm yr$^{-1}$ (0.43% yr$^{-1}$ and 0.46% yr$^{-1}$, respectively), and the trends in the magnitudes were closer to the observed midlatitude trend of 0.04 ppm yr$^{-1}$ (0.22% yr$^{-1}$) and subtropical trend of 0.05 ppm yr$^{-1}$ (0.61% yr$^{-1}$).

We note that the potential drivers of the amplitude increase during 1985–2013 were simulated to different levels of fidelity by the CESM: The 1985–2013 NH atmospheric temperature increase over land (1.02 K) was near the NCEP-NCAR Reanalysis (Kalnay et al., 1996) value (0.95 K), but the 2010 annual mean NH atmospheric $CO_2$ mole fraction in CESM was too high (425 ppm vs 391 ppm). Previous analysis of CESM shows that this high bias in simulated $CO_2$ is attributable to persistent
weak uptake in both land and ocean (Keppel-Aleks et al., 2013; Long et al., 2013).

## 3.2 Future $CO_2$ annual cycle amplitude changes

### 3.2.1 Total amplitude changes

Given the weak atmospheric $CO_2$ seasonal amplitude response in the CESM for the contemporary period, we examined the response of flux seasonality to stronger forcing in the FullyCoupled simulation run to 2300. Both near-surface atmospheric
temperature and the mean atmospheric $CO_2$ mole fraction increased markedly in the CESM (Fig. 5a, b). The accumulated mean NH atmospheric $CO_2$ mole fraction (Fig. 5a, solid black line) increased from approximately 320 ppm to 2350 ppm between 1950 and 2300. In the NoRad simulation, where radiative forcing was held fixed at 1850 levels, the atmospheric mole fraction followed a similar pattern of increase, but plateaued at a lower mean value by 2200. We note that the atmospheric $CO_2$ mole fraction values were diagnostic only, and the biogeochemical and radiative processes in the CESM responded to the
lower mole fraction values prescribed according to the ECP8.5 forcing scenario indicated by the dashed black line in Fig. 5a.

In the FullyCoupled simulation, the increases in $CO_2$ and other radiative forcing agents resulted in a 6 K temperature increase by 2100 and an 11 K temperature increase by 2300 relative to the 1950–1959 baseline mean (Fig. 5b). Temperature in the NoRad simulation only increased by ∼1.5 K through 2300, which can be traced to changes in albedo and surface energy balance. For small temperature changes and high levels of $CO_2$ fertilization, the NoRad experiment was able to maintain a
steady carbon sink of 4–5 Pg C yr$^{-1}$ between 2100 and 2300 (Fig. 5c). In contrast, the sink in the FullyCoupled simulation reached a maximum of 4.8 Pg C yr$^{-1}$ by 2120, then declined to 3–4 Pg C yr$^{-1}$ in the last 100 years of the simulation, suggesting that the extreme climate change in this simulation reduced the efficiency of the global terrestrial sink with time. LUC partly offset the weakening of the land carbon sink due to climate change (likely as a result of increased crop cover prescribed in the FullyCoupled simulation), enhancing net terrestrial carbon uptake by up to 2.8 Pg C yr$^{-1}$ after 2100.
The NH mean $CO_2$ annual cycle amplitude increased by 3.4 ppm (65%) by 2100, and 5.0 ppm (96%) by 2300 from the 1950–1959 baseline in the FullyCoupled simulation (Table 3). Consistent with the observed annual cycle amplification, the magnitudes of the $A^{FC}$ increases strengthened with increasing latitude (Fig. 7). $A^{FC}$ increases between 1950 and 2300 ranged from 1.4 ppm (56%) over the NH tropics to 10.6 ppm (122%) over the high latitudes (Table 3). Peak $A^{FC}$ magnitudes occurred between 2230 and 2250, and ranged from 4.2 ppm over the NH tropics to 19.5 ppm over the high latitudes (Table 4).
Given the weak baseline seasonal exchange in the CESM, simulated $CO_2$ annual cycle amplitudes did not approach the contemporary mean observed mean $A^{Obs}$ (2009–2013) until about 2240 in the NH high latitudes (Fig. 7a, black filled square, Table 4). Over the NH midlatitudes, subtropics, and tropics, peak $A^{FC}$ occurred by 2240, but values were still 0.4–2.5 ppm below current $A^{Obs}$. In the tropics, the discrepancy between $CO_2$ seasonality inferred only from simulated NEP and from

observations reflects the non-trivial contributions of ocean and fossil fuel fluxes to the $CO_2$ annual cycle (Randerson et al., 1997, Table 4) and (Lindsay et al., 2014, Fig. 15). When high latitude $CO_2$ amplitudes reached peak values, climate change and non-radiative forcing from $CO_2$ fertilization and N-deposition each contributed about half of the 9.6 ppm increase relative to the 1950–1959 baseline (Fig. 8). At this point, Arctic temperatures were 16 K higher than the baseline, midlatitude temperatures were 11 K higher (Fig. 7b, c), and the mean NH atmospheric $CO_2$ mole fraction was approximately 2330 ppm (Fig. 5a).

The relationship between the atmospheric $CO_2$ annual cycle amplitude and the NH annual net land carbon sink changed over the course of the FullyCoupled simulation. Several recent papers have hypothesized that the mean atmospheric $CO_2$ annual cycle may be a diagnostic of net terrestrial carbon uptake, since these variables tend to correlate positively in model simulations (e.g., Ito et al., 2016). While we found that the NH net land carbon sink and annual atmospheric $CO_2$ annual cycle amplitude were positively correlated through 2100 (Fig. 9), the net land carbon sink began to decrease before the $CO_2$ annual cycle amplitude decreased. In the FullyCoupled simulation, the NH decadal mean $CO_2$ annual cycle amplitude peaked near 10 ppm by 2240 (Table 4). However, the magnitude of the NH land sink began to decrease (Fig. 5c), likely reflecting enhanced subtropical respiration. This underscores that, while amplification of the $CO_2$ annual cycle may reflect enhanced seasonality of land carbon uptake, it does not necessitate enhanced annually-integrated land carbon uptake. Moreover, in the 23rd century, both the $CO_2$ annual cycle amplitude and the NH net land carbon sink declined, but there was evidence of hysteresis between the two quantities. In the latter part of the simulation when both the amplitude and net land carbon sink declined, the slope of their linear relationship was shallower than when both quantities were increasing before 2100. We hypothesize that the change in the relationship between the $CO_2$ annual cycle amplitude and the NH net land carbon sink resulted from respiration catching up to increased land carbon uptake after atmospheric $CO_2$ concentration, and thus the fertilization effect, leveled off in the last 100 years of the FullyCoupled simulation.

## 3.3 Contributions of non-radiative forcing, climate change, and LUC to amplitude trends

### 3.3.1 Effects of non-radiative forcing from $CO_2$ fertilization and N-deposition

In the CESM, $CO_2$ fertilization was the largest driver of $CO_2$ annual cycle amplification through 2300 over much of the NH (Fig. 8a), contributing 4.1 ppm to the 5.0 ppm increase in NH $A^{FC}$ ($\Delta A^{FC}$) between 1950 and the end of the 23rd century (Table 3). Results from Devaraju et al. (2016) suggest that global NPP is influenced equally by $CO_2$ fertilization and N-deposition over the historical period in the CESM. Therefore, trends in the $CO_2$ mean annual cycle amplitude likely responded to both drivers prior to 2100. While we cannot fully separate the influence of $CO_2$ fertilization and N-deposition given the experimental design, N-deposition was held fixed at 2100 values for the last 200 years of the simulations, so we expect that amplitude trends after 2100 mainly reflect enhanced $CO_2$ fertilization.

The amplitude increase owing to $CO_2$ fertilization and N-deposition (NoRad simulation) originated mainly from NH temperate regions (Fig. 10a), which accounted for 2.2 ppm (53%) of $\Delta A^{NoRad}$ at the end of the 23rd century (Table 3). Boreal regions (Fig. 11a) made the second greatest contribution (32%, 1.3 ppm) to NH $\Delta A^{NoRad}$. The remaining 15% (0.6 ppm) of the NH $\Delta A^{NoRad}$ increase came from the Arctic (Fig. 12a) or subtropical/tropical ecosystems. Temperate ecosystems had

the largest response to $CO_2$ fertilization and N-deposition. Consistent with this large temperate response, the increase in $A^{FC}$ from non-radiative forcing was largest over the NH midlatitudes (Table 3). The effects of $CO_2$ fertilization and N-deposition in boreal ecosystems contributed another 25% to the high latitude $CO_2$ amplitude, 22% to the midlatitude, and 60% to the subtropical $A^{FC}$.

In the CESM, the impact of $CO_2$ fertilization on the amplitude trend roughly scales with to the magnitude of overall GPP, consistent with hypotheses from Tans et al. (1990) and Schimel et al. (2015) that the fertilization effect on the land carbon sink is proportional to productivity. Thus, $CO_2$ fertilization and N-deposition effects on $A^{FC}$ were smallest in the Arctic, the region with the smallest GPP for the contemporary period. Fig. 13 shows that non-radiative forcing from boreal and temperate regions together constituted at least 35% of the increase in high latitude, midlatitude, and subtropical $A^{FC}$ from the beginning

of the 21st century through the end of the 23rd century. Furthermore, temperate $CO_2$ fertilization were the primary drivers of midlatitude $CO_2$ annual cycle amplification in all periods, and subtropical amplification from 2050 onward. In temperate and boreal regions, where $CO_2$ fertilization was the dominant driver of increases in the mean $CO_2$ annual cycle amplitude, increases in GPP seasonality outpaced increase in respiration. For example, in eastern temperate North America (ETNA), FullyCoupled GPP seasonal amplitudes increased from 6.8 Pg C in 1950 to 11 Pg C in 2250, while HR amplitudes increased from 0.85 Pg C

to 1 Pg C, and AR amplitudes increase from 4 Pg C to 7.6 Pg C. The strong fertilization effect on the amplification of the NH $CO_2$ annual cycle is surprising given that nitrogen limitation in the CLM4 produced weaker fertilization in the CESM compared to other CMIP5 models (Thornton and Zimmermann, 2007; Piao et al., 2013; Peng and Dan, 2015).

### 3.3.2    Climate change effects

During the early part of the simulation, boreal (Fig. 11b) and Arctic (Fig. 12b) climate change drove high-latitude atmo-

20 spheric $CO_2$ annual cycle amplification, increasing $A^{Clim}$ by 5 ppm by 2200 (Fig 13a). After 2200, $CO_2$ fertilization and climate change contributed nearly equally to high-latitude $CO_2$ annual cycle amplification. The increase in high latitude $A^{Clim}$ largely reflected warmer growing season temperatures that led to larger peak GPP values and a longer growing season. In the pulse regions that comprised the Arctic ecoclimate regions, annual mean GPP continued to increase with temperature ($r = 0.7 \, \mathrm{PgC} \, \mathrm{K}^{-1}, r^2 = 0.99$) until annual mean near-surface air temperatures surpassed 284 K.

By 2100, Arctic and boreal climate change increased high latitude $A^{FC}$ by about 1.3 and 2.5 ppm (39%) from the 1950–1959 mean, respectively, outweighing the combined contributions of boreal and temperate non-radiative forcing (2.2 ppm, 23%) (Fig. 13a). However, the effects of temperate and boreal $CO_2$ fertilization outweighed climate change effects on $CO_2$ annual cycle amplification at the end of the simulation. $CO_2$ fertilization and N-deposition added 5.3 ppm (62%) to high latitude, 6.8 ppm (81%) to midlatitude, and 3.5 ppm (69%) to the subtropical base period $A^{FC}$. It is worth noting that climate

change effects from lower latitudes made up most of the high latitude and subtropical residuals after 2000, and most of the midlatitude residual after 2200 (gray sections of bars in Fig. 13).

### 3.3.3 LUC effects

Land use change in the CESM decreased the seasonality of terrestrial $CO_2$ exchange (Figs. 8c–12c). Between 1850 (the year used for PFT fraction boundary conditions in the NoLUC simulation) and 2100, crop cover increased at the expense of grass and tree cover in the NH, reducing the seasonal amplitude of NH mean GPP for the duration of the simulation. As a result, LUC decreased the hemisphere-average atmospheric $CO_2$ annual cycle amplitude by 0.9 ppm (17%) from the 1950–1959 baseline in 2300 (Table 3).

This finding contrasts recent results suggesting that agriculture intensification contributes significantly to positive trends in the mean annual cycle amplitude (Gray et al., 2014; Zeng et al., 2014), and may reflect that croplands were treated as unmanaged grasslands in the CESM. Zeng et al. (2014) suggests that up to 45% of the observed trend may be due to land use practices, with the remainder partitioned roughly equally between climate and fertilization effects. We note that future LUC and land management will affect the $CO_2$ annual cycle, but that the overall impact the $CO_2$ annual cycle will depend on economic and social drivers in addition to climate feedbacks

### 3.3.4 Changes in growing season length

The growing season, defined as months with negative NEP (net terrestrial carbon uptake), increased for all NH terrestrial regions by about 1 month. The overall lengthened growing seasons accounted for 1–1.3% $yr^{-1}$ of the high latitude net terrestrial carbon uptake after 2050, and up to 5% $yr^{-1}$ of the midlatitude terrestrial carbon uptake after 2100. Thus, while this is an important contributor, it is secondary to increased mid-summer GPP.

The driver of the increased growing season length was different for different regions. For regions north of 30°N, climate change was the driver of increased growing season length, with boreal and temperate growing seasons increasing by one month after 2100 (Fig. 14a). Climate change extended the growing season for an additional month in the fall in the Arctic, and facilitated and earlier start to the midlatitude growing season in the spring (Fig. 14c).

$CO_2$ fertilization increased the growing season length in the subtropics (Fig. 14b) by increasing the water use efficiency (WUE, calculated as the ratio of canopy transpiration to GPP). For example, in subtropical Central America (Fig. 15), the period of net carbon uptake increased by one month on average in the NoRad simulation (Fig. 15a, magenta curve) In contrast, climate change altered the phasing of the mean annual cycle in the FullyCoupled simulation, with the growing season onset occurring earlier (Fig. 15a, black curve). GPP declined later in the FullyCoupled simulation despite WUE increases in the FullyCoupled simulation (Fig. 15b), suggesting that $CO_2$ fertilization effects were counteracted by climate change-driven reductions in total soil water (Fig. 15c, d) and increases in temperature in the FullyCoupled simulation. Conversely, during the latter part of the NoRad simulation, the growing season extended into June, for which WUE increased from around $2 \text{ g C kg}^{-1}\text{H}_2\text{O}$ in the 20th century to around $6 \text{ g C kg}^{-1}\text{H}_2\text{O}$ by 2300 and GPP likewise increased from 0.2 to 0.45 Pg C $\text{month}^{-1}$, despite essentially no change in soil water content (Fig. 15e, g). Increased GPP outweighed changes to respiration, and led to a shift toward net land carbon uptake during June (Fig. 15f, g).

## 4 Discussion and Conclusions

### 4.1 CO$_2$ annual cycle amplitude trends and applications for model evaluation

By analyzing CESM simulations run to 2300 with the ECP boundary conditions, we identified notable carbon cycle interactions that were not apparent before 2100, the nominal end date for CMIP5 runs. We found that the mean NH atmospheric CO$_2$ annual cycle amplitude increased by 65% from 1950 to 2100, and by an additional 30% by 2300 in the CESM1(BGC). Despite significant changes in climate and atmospheric CO$_2$ mole fractions in the extended simulation, the sensitivity of the CO$_2$ annual cycle amplitude to climate and fertilization drivers was remarkably similar both before and after 2100. We likewise found that regional contributions to the NH CO$_2$ seasonal amplitude trend were generally consistent throughout the simulation, in response to the second question posed in the introduction. CO$_2$ fertilization was the dominant driver of CO$_2$ annual cycle amplification for most of the NH, with the notable exception of Arctic ecosystems where temperature increases drove amplification prior to 2100. GPP leveled off in boreal high latitudes when annual mean temperatures exceeded 284 K, which contributed to small amplitude declines in the 23rd century. CO$_2$ fertilization increased the CO$_2$ annual cycle amplitude globally for the duration of the simulation even after the growth rate of CO$_2$ slowed during the 23rd century, suggesting that the CO$_2$ fertilization effect had not saturated in the CESM when CO$_2$ mole fractions were around 2000 ppm.

We saw evidence of hysteresis in the relationship between carbon cycle diagnostics that was only apparent after 2100, with respect to our third research question. For example, we found that the relationship between the NH mean CO$_2$ annual cycle amplitude and the NH net land carbon sink changed after 2100. A strong positive relationship between the CO$_2$ annual cycle amplitude and terrestrial carbon sink that has been noted previously by Ito et al. (2016) became decoupled in the mid-22nd century when sink strength began to decrease after 2100, while the CO$_2$ annual cycle amplitude continued to increase. When the NH CO$_2$ annual cycle amplitude began to decrease in the 23rd century, the correlation between the amplitude and land carbon sink weakened considerably compared to the relationship between the two quantities during the 20th and 21st century correlations.

Our analysis of extended concentration pathway simulations suggests several approaches for how to leverage multi-decadal atmospheric CO$_2$ observations for model evaluation. For example, consideration of only the trend in the hemispheric mean annual cycle, rather than trends in the latitudinally resolved mean annual cycle of CO$_2$, may obscure model biases. The Extended Concentration Pathway simulations run in the CESM with coupled biogeochemistry show that the NH mean annual cycle amplitude of atmospheric CO$_2$ increased by 16% from 1985 to 2013. The relative increase was in line with the observed 24% increase over the NH during the same time period. However, the spatial pattern of the relative amplitude change was more uniform throughout the NH than observed. Furthermore, the trend in the absolute magnitude of the amplitude at high latitudes was about half of the observed trend (0.05 ppm yr$^{-1}$ vs 0.09 ppm yr$^{-1}$). This result highlights the importance of considering meridionally-resolved atmospheric CO$_2$ data that explicitly accounts for the role of transport, since analysis of only hemispheric spatial patterns obscures incorrect spatial patterns simulated by the CESM. Analysis of the long-term CESM simulations are suggestive of patterns we might see in ongoing monitoring of the CO$_2$ annual cycle amplitude. CESM simulations show that the major drivers of the mean annual cycle amplification impact differential imprints on atmospheric

$CO_2$ in different latitude bands. For example, $CO_2$ fertilization leaves the largest imprint in both absolute and relative terms on midlatitude $CO_2$, whereas climate change may amplify high latitude $CO_2$ while having a near-neutral impact on $CO_2$ annual cycle amplitudes south of 60°N (Fig. 13). These fingerprints may be useful for developing hypotheses regarding observed trends and determining future observational strategies to monitor carbon-climate feedbacks.

Several recent papers have considered how the amplitude of NH net carbon exchange has changed over the historical period in different categories of prognostic models. Ito et al. (2016) analyze MsTMIP terrestrial ecosystem models to determine how atmospheric $CO_2$, climate change, and land use affect the NH flux amplitude for the historical period, and Zhao et al. (2016) analyze the net terrestrial flux to the atmosphere in TRENDY models. Both of these studies find that $CO_2$ fertilization is the strongest driver of increasing ecosystem productivity and thus the amplitude of the net carbon exchange in the NH, consistent
with our results. A significant difference between the approach used by these papers and our study is that they consider the net flux amplitude, whereas we propagate fluxes using an atmospheric transport operator to determine the influence on latitudinally-resolved atmospheric $CO_2$ fields. Given the importance of atmospheric transport on the mean annual cycle of atmospheric $CO_2$ (e.g., Barnes et al., 2016), and the small biases induced by the simplified pulse-response transport operator, we recommend that future studies explicitly simulate the full atmospheric $CO_2$ field.

**4.2   Uncertainties and future model needs**

The mean annual cycle of atmospheric $CO_2$ is a first-order diagnostic of terrestrial carbon exchange and its trend with time integrates a range of environmental and human factors (Randerson et al., 1997). An active area of carbon cycle research is determining the extent to which coupled ESMs provide predictive skill for future carbon-climate feedbacks. We note that many of the methods used to evaluate the carbon cycle in ESMs rely on benchmarking short-term responses to either seasonal or
interannual climate variability (Keppel-Aleks et al., 2013), or on extrapolating future behavior based on some mechanistic link between short-term and long-term variability (Cox et al., 2013; Hoffman et al., 2014). The changing $CO_2$ annual cycle provides a unique opportunity to gauge a model's sensitivity to slow-varying climate and environmental changes, since we have observed large trends in this quantity over the instrumental record (Graven et al., 2013).

However, biases in seasonality in the CESM1(BGC) that lead to smaller increases in NH atmospheric $CO_2$ seasonal ampli-
tudes in the CESM compared to observations during 1950–2010 prompt further model development. Moreover, the relatively smooth response to extreme changes in temperature and $CO_2$ suggests that the CESM may not parameterize processes that could cause nonlinear carbon cycle feedbacks. CESM1(BGC) did not include paramaterizations for permafrost carbon dynamics, which have since been improved in CESM (Koven et al., 2015). The lack of permafrost dynamics likely has a large impact on $CO_2$ annual cycle trends, especially later in the simulation when global mean temperature has increased by over 10 K in
the FullyCoupled simulation. Thus, short soil carbon turnover time in CLM4 may have contributed to the amplitude underestimation by damping ecosystem respiration outside of the growing season (Keppel-Aleks et al., 2013; Koven et al., 2013) and would affect both baseline values and trends. Ongoing model development in the CESM includes improved representation of permafrost carbon (Koven et al., 2015), and thus future model configurations will provide an improved tool for investigating a process that may provide one of the tipping points we identified in our key science questions.

In addition, Forkel et al. (2016) found that interaction between climate change and changes in vegetation cover over northern high latitudes was the primary driver of the north-south gradient in observed NH atmospheric $CO_2$ seasonal amplitude trends, indicating that the lack of dynamic vegetation in the CLM4 likely contributes to underestimation of the seasonal amplitudes by the CESM. Tree cover is expected to expand further northward with climate change (e.g., Lloyd, 2005), which may contribute to the long-term increase in NEP flux amplitude within high latitude ecosystems. In contrast, drying at lower latitudes may lead to replacement of trees with grasses and subsequent decreases in NEP amplitude. An ecosystem demography version (CLM-ED) that will permit successional patterns in response to environmental change is presently under development. We consider the documentation of trends in the static-vegetation configuration presented in this manuscript to be a crucial first step toward eventually determining the sensitivity of land-atmosphere biogeochemical couplings in more sophisticated future configurations of the CESM model.

Development is also under way to represent irrigation and fertilization in croplands in future versions of the CLM. Gray et al. (2014) and Zeng et al. (2014) suggest that agricultural amplification, facilitated by irrigation and fertilization, may be an important driver of the observed mean annual cycle trend. In the CESM, however, crop cover is currently treated as unmanaged grass. Thus, these agricultural practices are not explicitly modeled, and do not mitigate the reduction in tree cover in the FullyCoupled simulation. Our results indicate that explicit consideration of human modifications may be necessary for prognostic models both to match observations and to provide realistic predictions of future changes. After accounting for land management contributions to the amplitude increase, the sensitivity of the $CO_2$ amplitude to natural factors may be reasonable. Our results suggest that model development focused on human modification of carbon fluxes (e.g., by agriculture (Levis et al., 2014) or by disturbance (Kloster et al., 2012) may facilitate improved comparison both of mean behavior and trends.

*Acknowledgements.* This research was supported in part by the Biogeochemistry – Climate Feedbacks Scientific Focus Area (SFA), which is sponsored by the Regional and Global Climate Modeling (RGCM) Program in the Climate and Environmental Sciences Division (CESD) of the Biological and Environmental Research Program in the U.S. Department of Energy Office of Science.

The National Center for Atmospheric Research (NCAR) is sponsored by the National Science Foundation.

Computing resources were provided by the Climate Simulation Laboratory at NCAR's Computational and Information Systems Laboratory (CISL), sponsored by the National Science Foundation and other agencies.

We gratefully acknowledge Ernesto Munoz for running and processing the CESM simulations.

We thank two anonymous reviewers for their thoughtful comments that greatly improved the manuscript.

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

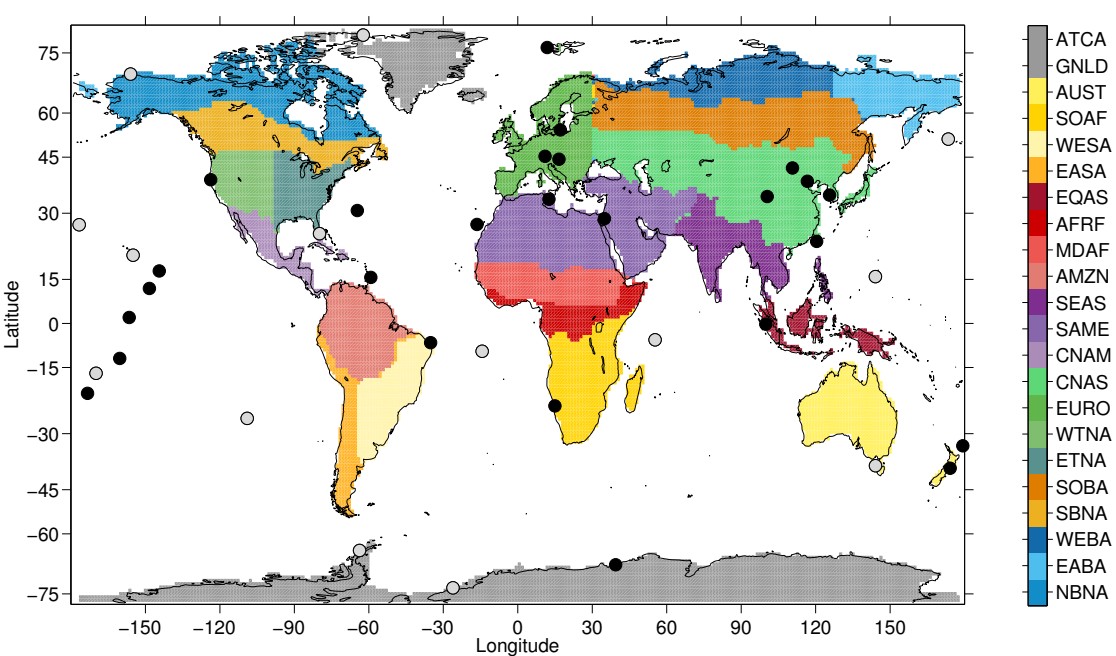

**Figure 1.** Map of stations where $CO_2$ from the pulse response code is computed. Table 1 lists the station identifiers and locations. The GEOS-Chem regions defined in the pulse response code (see Table 2) are shaded and grouped according the Arctic (blue shades), boreal (orange shades), NH temperate (green shades), NH subtropical (purple shades), tropical (red shades), and SH land (yellow shades) ecoclimate regions. Greenland and Antarctica (gray shades) were excluded from the analysis. Gray circles indicate a subset of stations from which the atmospheric annual $CO_2$ amplitudes were computed from 1985–2013 monthly mean observations (bold stations in Table 1).

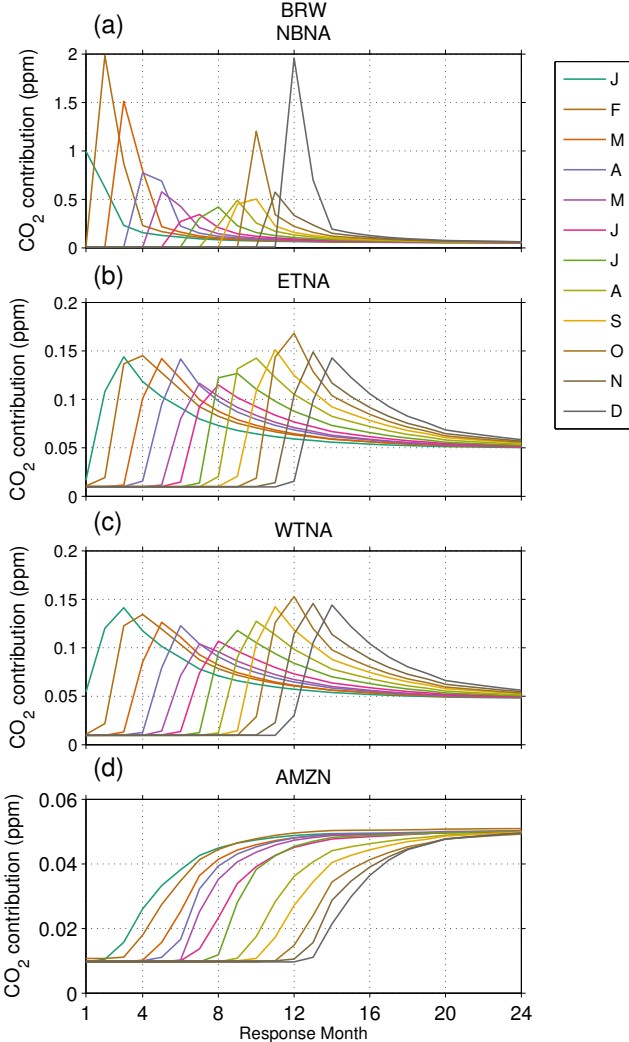

**Figure 2.** The imprints of 1 PgC pulses emitted in individual months (colored curves) on atmospheric $CO_2$ at Barrow (BRW). Imprints are shown from selected regions: (a) Northern Boreal North America (NBNA, (b) Eastern Temperate North America (ETNA), (c) Western Temperate North America (WTNA), and (d) Amazon (AMZN) on the atmosphere sampled at Barrow (BRW). For clarity, we plot only the first calendar year (months 13–24) after the pulses were released.

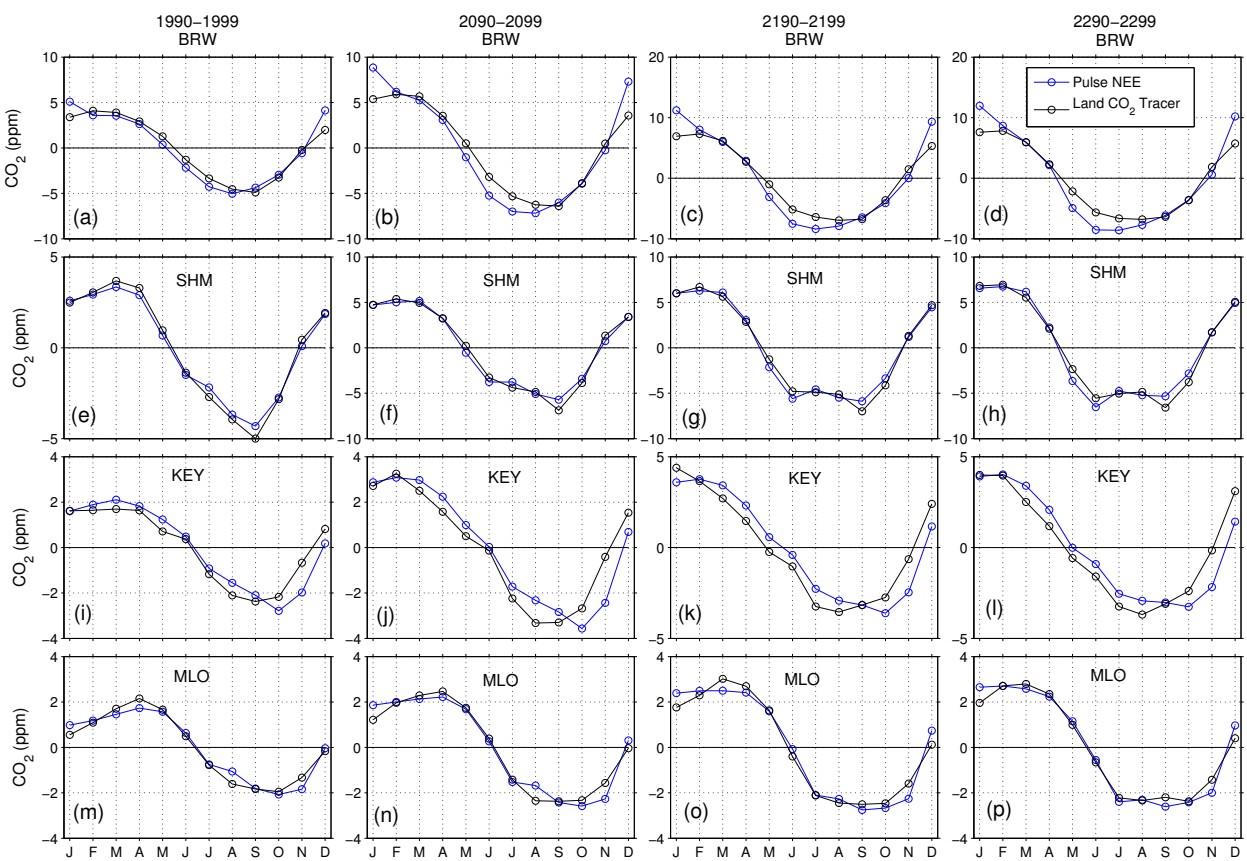

**Figure 3.** Mean annual cycles of atmospheric $CO_2$ derived from (blue curves) NEE run through the pulse-response function and (black curves) the CESM land $CO_2$ tracer for (a–d) Barrow (BRW), (e–h) Shemya Island (SHM), (i–l) Key Biscayne (KEY), and (m–p) Mauna Loa (MLO) in 1990–1999, 2090–2099, 2190–2199, and 2290–2299.

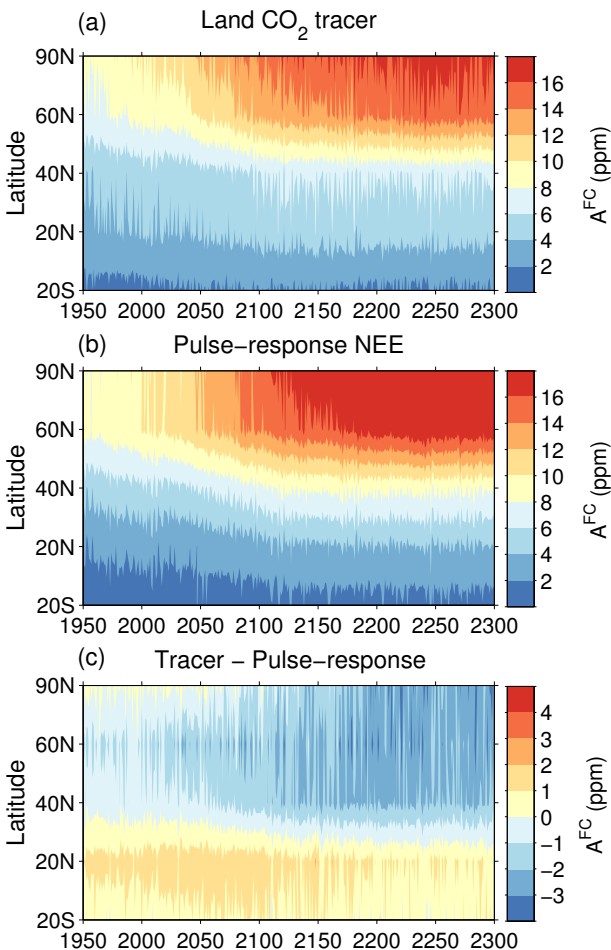

**Figure 4.** $CO_2$ annual cycle amplitudes in the FullyCoupled simulation derived from (a) the CESM land $CO_2$ tracer, and (b) running NEE from the CLM4 through the pulse-response code. (c) The difference between the land tracer and NEE-derived $CO_2$ annual cycle amplitudes.

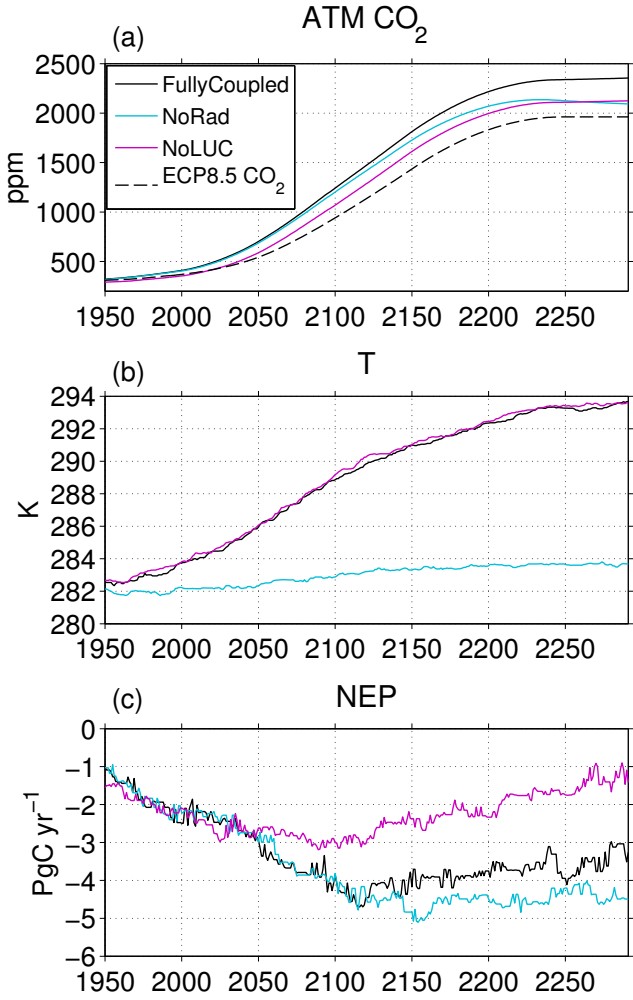

**Figure 5.** (a) Annual mean atmospheric $CO_2$, (b) annual mean bottom-level atmospheric T over land, and (c) annual mean NEP averaged over the NH ($0°$–$90°$N) in the CESM FullyCoupled (solid black curves), NoRad (blue curves), and NoLUC (magenta curves) simulations. Negative NEP indicates net annual $CO_2$ uptake by the land surface. Values are filtered using a 10-year running median. The $CO_2$ mole fraction values in (a) result from the net contributions of land, ocean, and fossil fuel tracers calculated from NEE as described in Section 2.3, and differ from the atmospheric $CO_2$ mixing ratio that is prescribed according to the ECP8.5 scenario (dashed black curve).

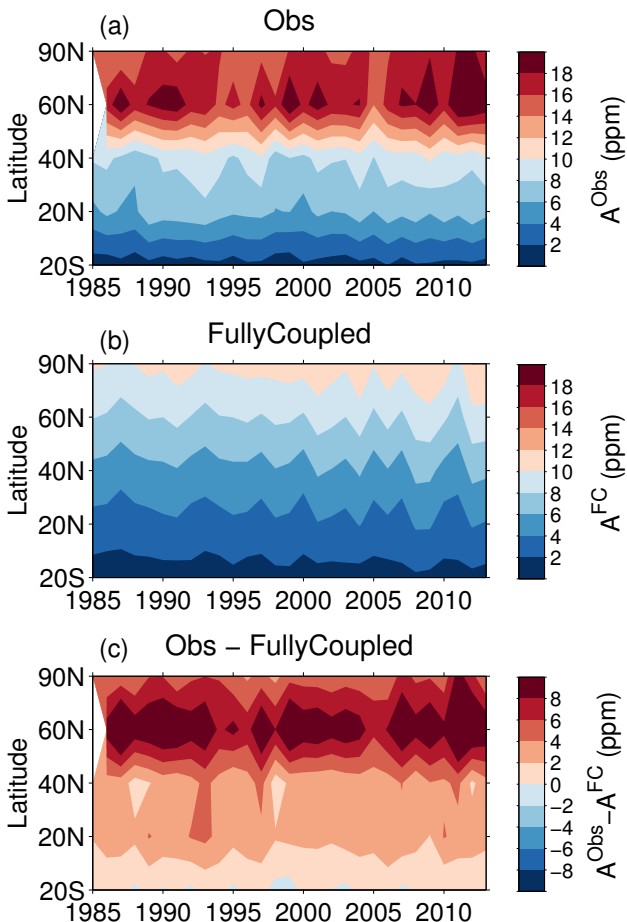

**Figure 6.** (a) Observed atmospheric $CO_2$ annual cycle amplitudes ($A^{Obs}$), (b) FullyCoupled atmospheric $CO_2$ annual cycle amplitudes ($A^{FC}$), and (c) the difference between $A^{Obs}$ and $A^{FC}$ during 1985–2013.

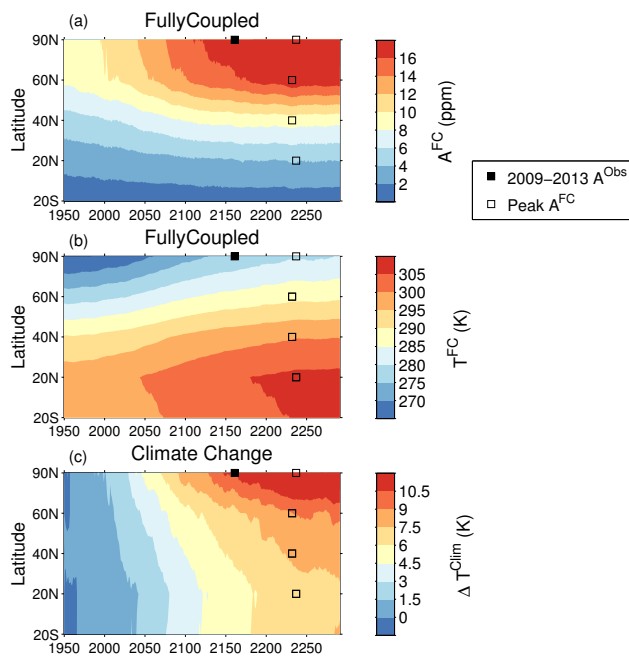

**Figure 7.** (a) 10-year moving averages of FullyCoupled atmospheric $CO_2$ annual cycle amplitude ($A^{FC}$) from 1950–1959 to 2291–2300. The black square indicates the decade in which the CESM reached observed $CO_2$ annual cycle amplitudes averaged over 2009–2013 in the NH high latitudes (Table 4). Open squares indicate the decades when peak amplitudes occurred. In the NH midlatitudes, subtropics, and tropics, peak amplitudes did not reach present observed values. (b) Atmospheric temperatures over land in the FullyCoupled simulation ($T^{FC}$). (c) The change in atmospheric temperature with respect to the 1950–1959 mean caused by climate change ($\Delta T^{Clim}$).

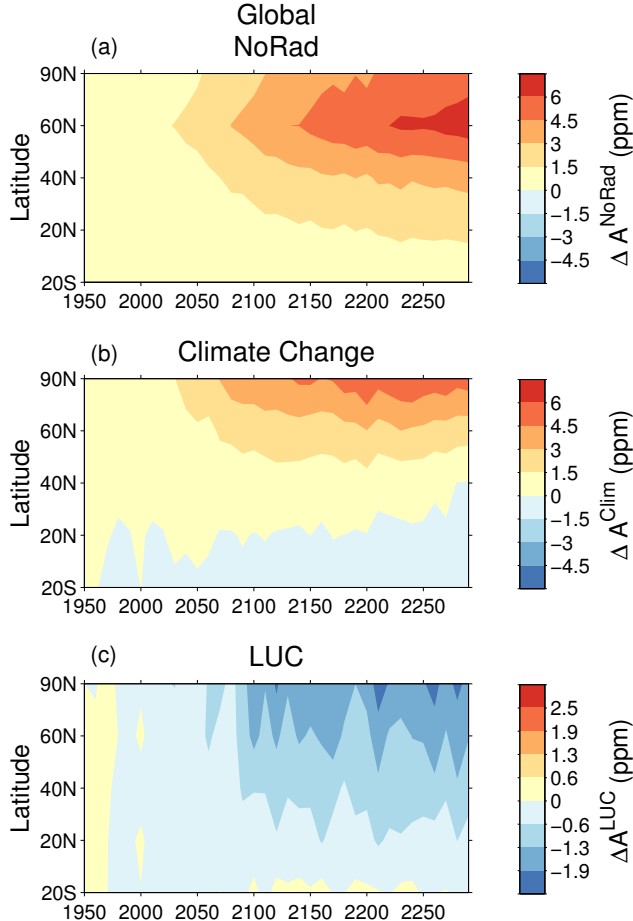

**Figure 8.** Changes in the decadal mean atmospheric $CO_2$ annual cycle amplitudes from 1950–1959 values from (a) $CO_2$ fertilization and N-deposition ($\Delta A^{NoRad}$), (b) climate change ($\Delta A^{Clim}$), and (c) land use change ($\Delta A^{LUC}$) from all land regions in each latitude bin. Note that the color scale in (c) is different from the scale in (a) and (b).

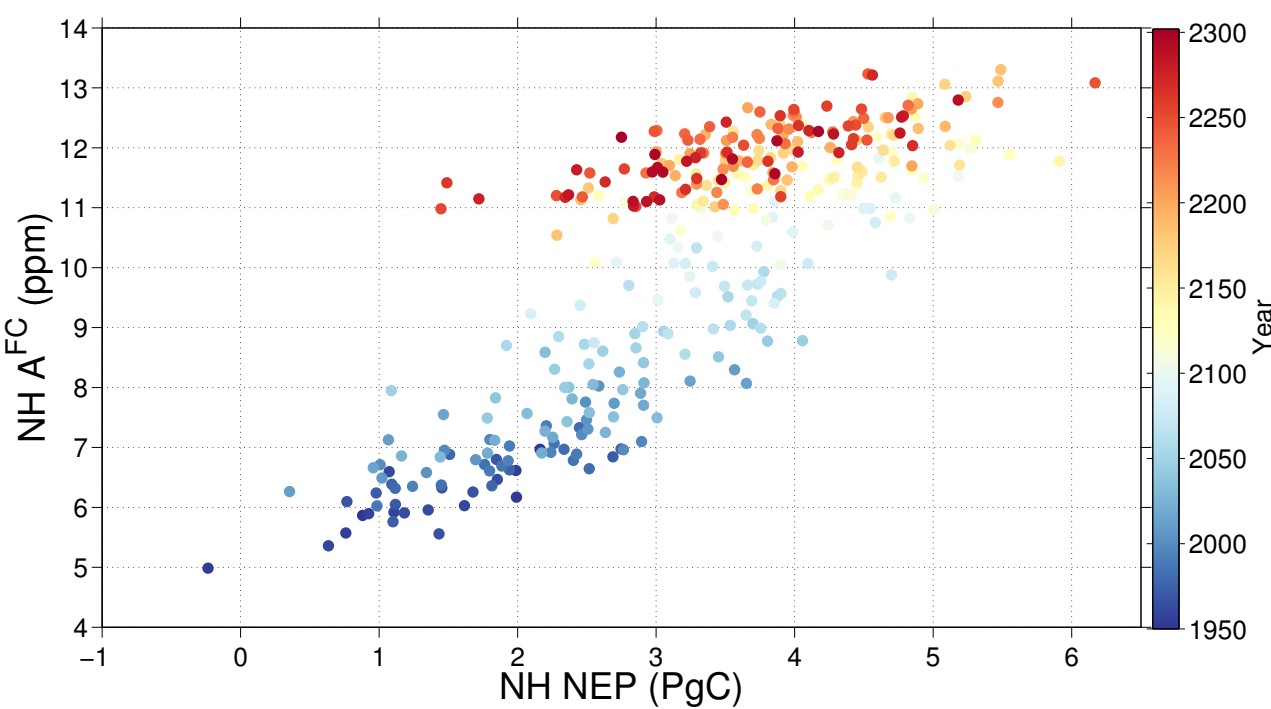

**Figure 9.** NH mean $CO_2$ annual cycle amplitude (ppm) as a function of annual mean NH NEP (PgC). NEP < 0 indicates net carbon uptake by land, and circles are shaded by simulation year. The relationship between amplitude and net land carbon exchange shows evidence of hysteresis, with different slopes before and after 2100.

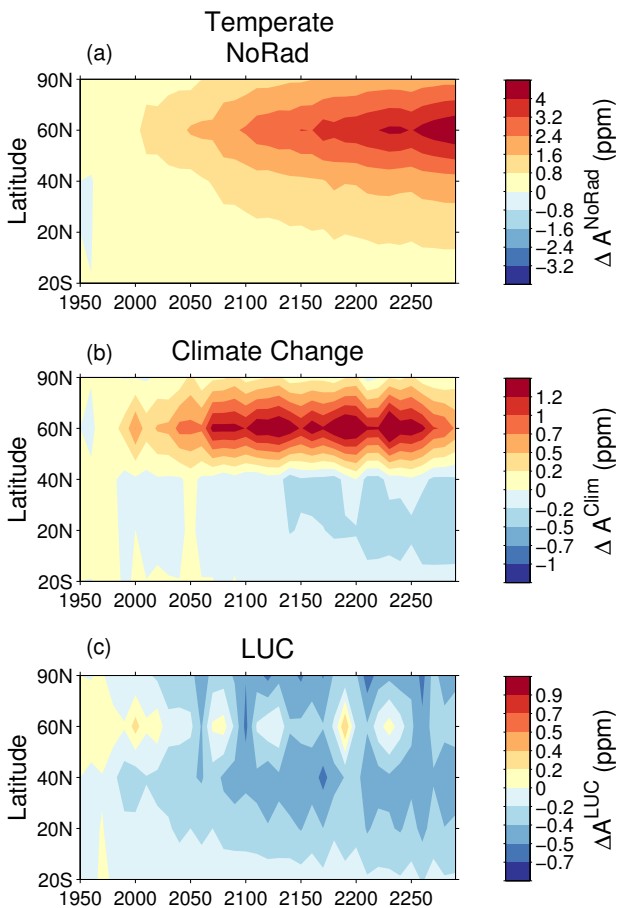

**Figure 10.** As in Fig. 8 but for (a) $CO_2$ fertilization + N-deposition, (b) climate change, and (c) LUC from NH temperate regions. Note that the color scales are different in each panel.

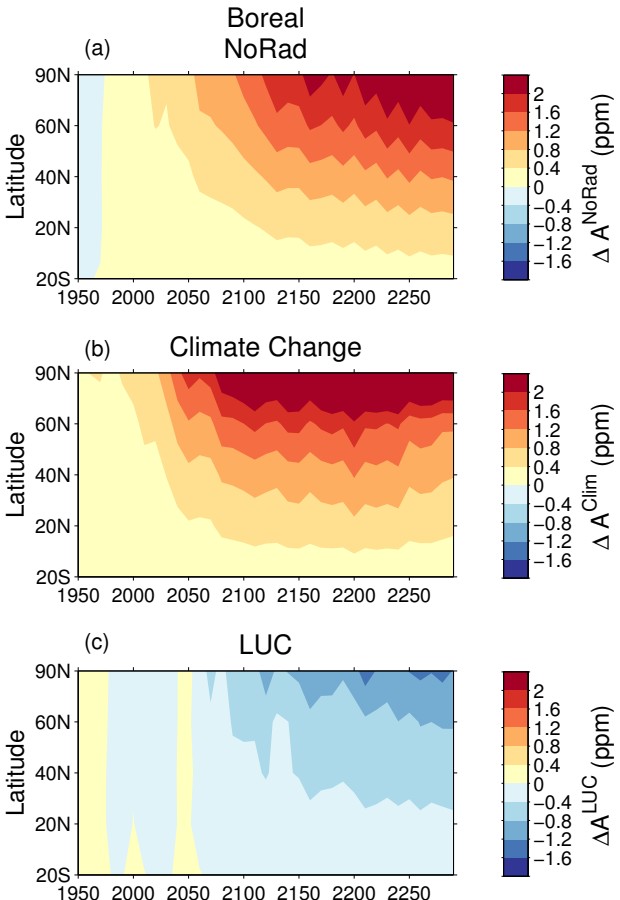

**Figure 11.** As in Fig. 8 but for (a) $CO_2$ fertilization + N-deposition, (b) climate change, and (c) LUC from boreal regions.

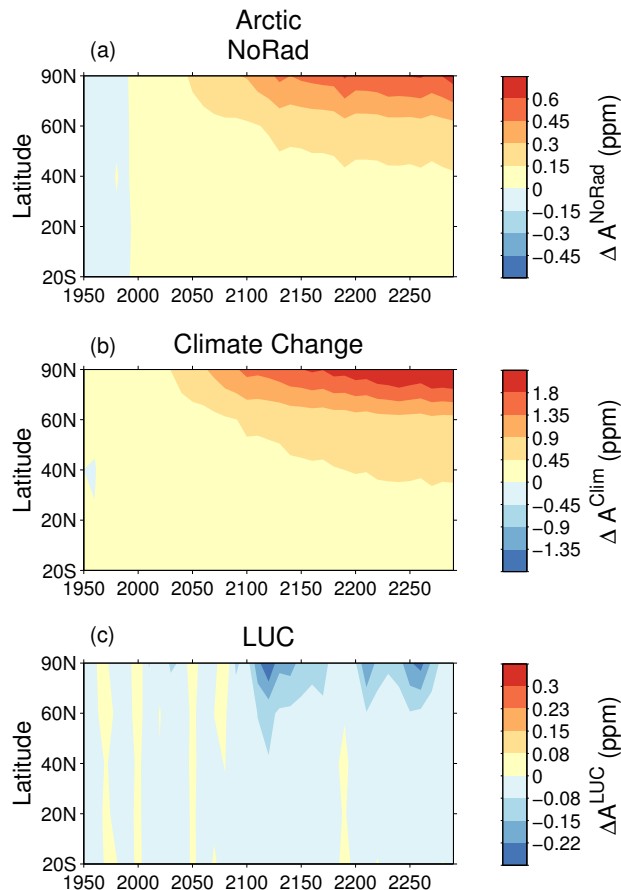

**Figure 12.** As in Fig. 8 but for (a) $CO_2$ fertilization + N-deposition, (b) climate change, and (c) LUC from the Arctic. Note that the color scales are different in each panel.

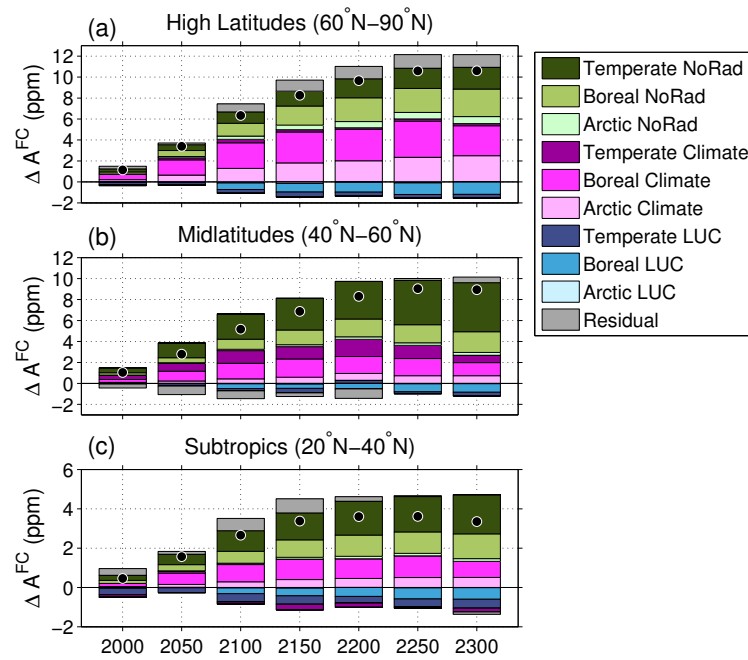

**Figure 13.** Contributions of temperate, boreal, and Arctic combined $CO_2$ fertilization and N-deposition (NoRad), climate change (Climate), land use change (LUC) to the change in the FullyCoupled mean atmospheric $CO_2$ annual cycle amplitude from 1950–1959 values ($\Delta A^{FC}$) in the decades ending in 2000, 2050, 2100, 2150, 2200, 2250, and 2300 over (a) the NH high latitudes, (b) the NH midlatitudes, and (c) the NH subtropics. Residual values are the sums of the contributions of LUC, climate change, $CO_2$ fertilization, and N-deposition in the subtropical, tropical, and SH GEOS-Chem land regions. Negative values indicate that the forcing decreased the atmospheric $CO_2$ annual cycle amplitude. Units are ppm.

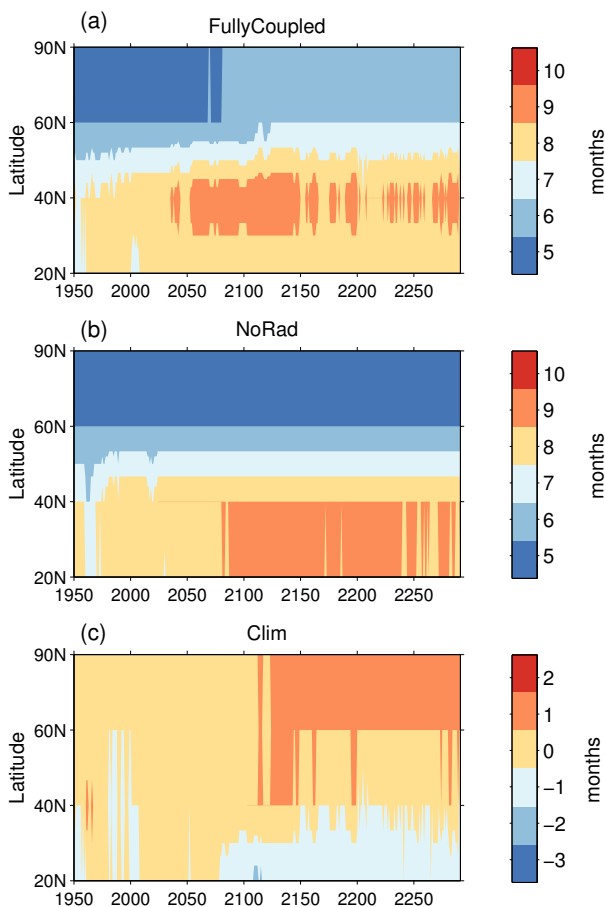

**Figure 14.** Growing season length (10-year running median of months with NEP < 0) in each latitude band in the (a) FullyCoupled, and (b) NoRad simulations. (c) The contribution of climate change to growing season length.

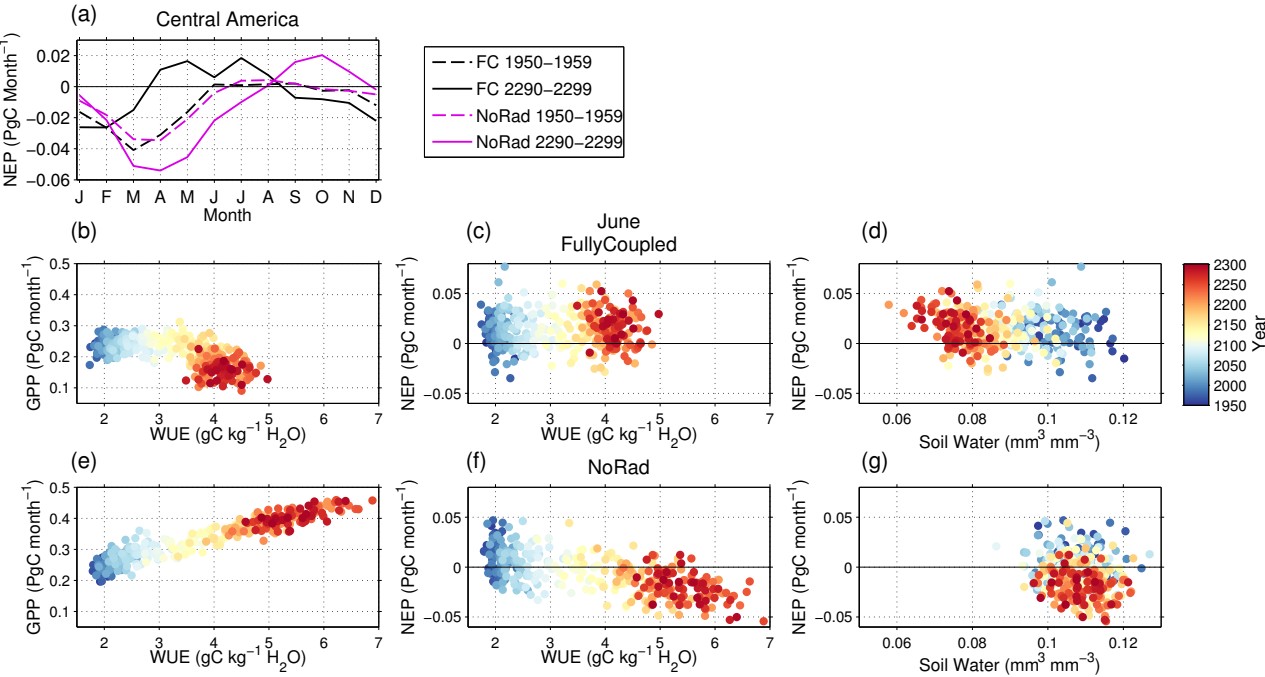

**Figure 15.** (a) Mean annual cycles of NEP in Central America averaged over 1950–1959 and 2290–2299 in the FullyCoupled (black dashed and solid lines) and NoRad (magenta dashed and solid lines) simulations. (b, e) GPP as a function of WUE, (c, f) NEP as a function of WUE, and (d, g) NEP as a function of volumetric soil water in the top layer (0.71 cm depth) in the month of June over Central America in the (b–d) FullyCoupled simulation. (e–g) the NoRad simulation. NEP < 0 indicates net carbon uptake by land.

**Table 1.** Latitude band, station location, station ID, latitude, and longitude of $CO_2$ sample locations. Observations from bold stations were analyzed over the 1985–2013 period.

| Latitude Band | Station Location | Station ID | Latitude | Longitude |
|---|---|---|---|---|
| NH High Latitudes | **Alert, Nunavut** | **ALT** | **82.45** | **297.49** |
| | Ny-Alesund, Svalbard | ZEP | 78.90 | 11.90 |
| | **Barrow, Alaska** | **BRW** | **71.30** | **203.40** |
| NH Midlatitudes | Baltic Sea | BAL | 55.35 | 17.22 |
| | **Shemya Island, Alaska** | **SHM** | **52.70** | **174.10** |
| | Hohenpeissenberg, Germany | HPB | 47.80 | 11.02 |
| | Hegyhatsal, Hungary | HUN | 46.95 | 16.65 |
| | Ulaan Uul, Mongolia | UUM | 44.45 | 111.10 |
| | Trinidad Head, California | THD | 41.10 | 235.80 |
| | Shangdianzi, China | SDZ | 40.65 | 117.12 |
| NH Subtropics | Tae-ahn Peninsula, Rep. Korea | TAP | 36.70 | 126.10 |
| | Mt. Waliguan, China | WLG | 36.29 | 100.90 |
| | Lampedusa, Italy | LMP | 35.52 | 12.62 |
| | Tudor Hill, Bermuda | BMW | 32.30 | 295.10 |
| | WIS Station, Negev Desert, Israel | WIS | 29.97 | 35.06 |
| | Izana, Tenerife, Canary Islands | IZO | 28.31 | 343.50 |
| | **Sand Island, Midway, USA** | **MID** | **28.21** | **182.62** |
| | **Key Biscayne, Florida** | **KEY** | **25.67** | **279.84** |
| | Lulin, Taiwan | LLN | 23.47 | 120.87 |
| NH Tropics | **Mauna Loa, Hawaii** | **MLO** | **19.50** | **204.42** |
| | Pacific Ocean (15°N) | POCN15 | 15.00 | 215.00 |
| | **Mariana Islands, Guam** | **GMI** | **13.39** | **144.66** |
| | Ragged Point, Barabados | RPB | 13.17 | 300.57 |
| | Pacific Ocean (10°N) | POCN10 | 10.00 | 211.00 |
| | **Christmas Island, Rep. Kiribati** | **CHR** | **1.700** | **202.85** |
| SH Tropics | Bukit Kototabang, Indonesia | BKT | -0.20 | 100.32 |
| | **Mahe Island, Seychelles** | **SEY** | **-4.68** | **55.53** |
| | Maxaranguape, Brazil | NAT | -5.52 | 324.74 |
| | **Ascension Island, UK** | **ASC** | **-7.97** | **345.60** |
| | Pacific Ocean (10°S) | POCS10 | -10.00 | 199.00 |
| | Tutuila, American Samoa | SMO | -14.25 | 189.44 |
| | Pacific Ocean (20°S) | POCS20 | -20.00 | 186.00 |
| SH | Gobabeb, Namibia | NMB | -23.58 | 15.03 |
| | **Easter Island, Chile** | **EIC** | **-27.16** | **250.57** |
| | Pacific Ocean (35°S) | POCS35 | -35.00 | 180.00 |
| | **Cape Grim, Tasmania, Australia** | **CGO** | **-40.68** | **144.69** |
| | Baring Head Station, NZ | BHD | -41.41 | 174.87 |
| | **Palmer Station, Antarctica** | **PSA** | **-64.92** | **296.00** |
| | Syowa Station, Antarctica | SYO | -69.01 | 39.59 |
| | **Halley Station, Antarctica** | **HBA** | **-75.61** | **333.79** |
| | South Pole, Antarctica | SPO | -89.90 | 335.20 |

**Table 2.** GEOS-Chem regions that comprise the ecoclimate regions shown in Fig. 1.

| Ecoclimate Region | Pulse Region | Acronym |
|---|---|---|
| Arctic | Northern Boreal North America | NBNA |
| | Eastern Boreal Asia | EABA |
| | Western Boreal Asia | WEBA |
| Boreal | Southern Boreal North America | SBNA |
| | Southern Boreal Asia | SOBA |
| NH Temperate | Eastern Temperate North America | ETNA |
| | Western Temperate North America | WTNA |
| | Europe | EURO |
| | Central Asia | CNAS |
| NH Subtropics | Central America | CNAM |
| | Sahara and Middle East | SAME |
| | Southeast Asia | SEAS |
| NH Tropics | Amazon | AMZN |
| | Middle Africa | MDAF |
| | African Rainforest | AFRF |
| | Equatorial Asia | EQAS |
| SH Land | Eastern South America | EASA |
| | Western South America | WESA |
| | South Africa | SOAF |
| | Australia | AUST |
| Polar | Greenland | GNLD |
| | Antarctica | ATCA |

**Table 3.** Cumulative changes in mean $CO_2$ annual cycle amplitudes (ppm) from the 1950–1959 baseline in the FullyCoupled simulation, and the contributions of non-radiative forcing ($CO_2$ fertilization and N-deposition), climate change, and LUC to the FullyCoupled amplitude changes averaged over the NH high latitudes, midlatitudes, subtropics, tropics, and the NH.

| | FullyCoupled | | | NoRad | | | Climate Change | | | LUC | | |
|---|---|---|---|---|---|---|---|---|---|---|---|---|
| Year | 2100 | 2200 | 2300 | 2100 | 2200 | 2300 | 2100 | 2200 | 2300 | 2100 | 2200 | 2300 |
| High Latitudes | 6.4 | 9.6 | 10.6 | 2.4 | 4.5 | 5.3 | 4.0 | 5.1 | 5.2 | -1.1 | -1.3 | -1.7 |
| Midlatitudes | 5.2 | 8.3 | 8.9 | 3.3 | 5.7 | 6.8 | 1.9 | 2.6 | 2.1 | -0.9 | -0.7 | -1.3 |
| Subtropics | 2.8 | 3.6 | 3.4 | 1.8 | 3.0 | 3.5 | 1.0 | 0.6 | -0.1 | -0.7 | -0.9 | -0.9 |
| NH Tropics | 1.0 | 1.5 | 1.4 | 0.9 | 1.5 | 1.8 | -0.1 | -0.1 | -0.3 | -0.5 | -0.4 | -0.4 |
| NH | 3.4 | 4.7 | 5.0 | 2.0 | 3.4 | 4.1 | 1.4 | 1.3 | 1.0 | -0.6 | -0.7 | -0.9 |

**Table 4.** Observed and FullyCoupled 2009–2013 mean $CO_2$ annual cycle amplitudes, and corresponding decades when peak FullyCoupled amplitudes occurred in each latitude band shown in Fig. 6a.

| Latitude | $A^{Obs}$ (ppm) 2009–2013 | $A^{FC}$ (ppm) 2009–2013 | Mean $A^{FC}$ (ppm) during peak decade | Center year of Decade of Peak $A^{FC}$ |
|---|---|---|---|---|
| High Latitude | 17.3 | 11.3 | 19.5 | 2241 |
| Midlatitude | 19.5 | 8.7 | 17.8 | 2236 |
| Subtropics | 9.3 | 5.9 | 9.0 | 2236 |
| NH Tropics | 6.7 | 3.7 | 4.2 | 2241 |
| NH mean | 11.3 | 6.2 | 10.4 | 2236 |