# Peer review of "Drivers of Multicentury Trends in the Atmospheric CO2 Mean Annual Cycle in a Prognostic ESM"

_Biogeosciences, 2016_

## Referee Comment (RC1) · Anonymous Referee #1 · 17 Jun 2016

This study explores different contributors to the increase in the atmospheric CO2 seasonal amplitude, as predicted by the CESM in simulations that span 1950-2300. I am generally supportive of this paper. Clearly an impressive effort went into it and it is well organized and written. However, I have some major concerns, listed in order of decreasing priority:

1) There are some steps in the methodology that need more detail and justification – some of them could/should really be stand-alone papers. These include a) The pulse-response method. b) The documentation of mid-latitude trends in observed CO2 amplitude

2) The CESM does a poor job of reproducing the current CO2 amplitude and the historical observed amplitude trends, which undermines confidence in the results presented

here. Although I think the exercise is still worthwhile, some sort of well thought out rationale or statement is needed to explain why readers should believe or pay any heed to the future model results going out to 2300, e.g., are there certain results that are robust and insightful despite the model's poor present-day performance?

Expanding on 1a) The pulse-response method. This could really be a stand-alone paper (see, e.g., Nevison, C.D., D.F. Baker, and K.R. Gurney, A methodology for estimating seasonal cycles of atmospheric CO2 resulting from terrestrial net ecosystem exchange (NEE) fluxes using the Transcom T3L2 pulse-response functions, Geosci. Model Dev. Discuss., 5, 2789-2809, 2012, www.geosci-model-dev-discuss.net/5/2789/2012/ doi:10.5194/gmdd-5-2789-2012, 2012.)

While I support the method and realize that it would be prohibitively expensive computationally to break down the contributions to CO2 amplitude change from different regions and mechanisms without some sort of shortcut approach like the Pulse Response method, I think it needs more than a 1 paragraph explanation. For example:

i) Is there any IAV in the meteorology used to create the pulse fields? Also, what is the consequence of assuming those met fields will still apply in 2300? ii) How are the 60-month decaying pulses combined to create a model atmospheric CO2 cycle? iii) In figure 2, the pulse-response amplitudes at midlatitudes are 3ppm or more smaller than the fully prognostic tracer. This doesn't seem "broadly similar" and undermines confidence that this methodology can detect subtle trends, esp. in the midlatitudes. iv) The GMD Discussions paper above was never accepted for final publication, due to reviewers who thought adjoint methods were superior. While the current method is superior in that it divides land into a larger number of regions (20 v. 11), the GMDD paper on the other hand was applying the method to estimate mean seasonal cycles, which are easier to get right than the more subtle trends in amplitude over time examined here.

Expanding on 1b) I'm not sure there is any evidence that CO2 seasonal amplitude is increasing at midlatitude sites such as NWR or UUM, KZM/D. In fact, if anything, they

may be decreasing – possibly due to drought effects. The most robust effects are seen at BRW, with the amplitude increase at MLO less than half that of BRW. I don't think Zeng et al., 2014 is an adequate reference to prove that midlatitude $CO_2$ amplitude is increasing, since they don't actually show this.

Minor comments: p.1,L8, The term "changing atmospheric composition" to encompass $CO_2$ fertilization and N deposition is confusing. These two don't really belong in the same category, in my opinion, since the N deposition is relevant mainly after it deposits on the soil, i.e., the authors are not looking at some sort of physiological response of plants to increased atmosphere NOx or NH3 concentration. p.1,L12 is confusing as written – in one case we have the end time (2300) and in the other we have the start time (after 2100). Please rewrite to clarify start and end times for both effects p.1,L15 "rather than the strength of the terrestrial carbon sink" please explain more clearly what is meant here. p.1,L17, suggest replacing "is not predicated on" with "does not necessarily imply" p.1, L20 I think it's more accurate to say "at some NH sites" rather than "over the NH" (see my comments above about midlatitude trends). p.2,L31 missing AND between citations. p.2, L35 suggest saying, "Model evidence suggests that the combined effects . . ." and delete "in simulations." P2.,L20 and p.3,L17 again I find the catch-all term "changes in atmospheric composition" confusing. p. 6, L30. It seems like a stretch to call 425 ppm and 391 ppm "roughly equivalent" p.7,L5 Please provide a reference for the observed mid-latitude trend of 0.04 ppm yr-1. P8, L19 Please explain further. Why is this consistent with effects being proportional to GPP? P9L12, to avoid confusion, would suggest splitting into 2 sentences: "..simulation. These latter influences added 4.7 ppm . . ." P9, L27 The Zeng et al reference, in my reading, does not actually demonstrate that the spatial distribution of where atmospheric $CO_2$ amplitude increases are seen (mainly at high latitudes) are consistent with agriculture, which is large at mid-latitudes. P10, L23, "perhaps indicating . . ." Please explain further.

---

## Referee Comment (RC2) · Anonymous Referee #2 · 11 Jul 2016

The paper describes an analysis of potential drivers of multi-century trends in the seasonal cycle amplitude of the atmospheric CO2 concentration with a Prognostic Earth System Model. The study follows from the paper of Graven et al. 2013 that analyzed in detail the large increase of atmospheric CO2 seasonal cycle amplitude at high northern latitudes over the pas 60 years; In a series of studies trying to disentangle the drivers of the observed increase in atmospheric CO2 seasonal amplitude, this paper propose a first attempt with a prognostic coupled carbon-climate cycle model and an investigation of the amplitude changes up to the horizon 2300.

The paper is clearly written and relatively easy to follow. However it seems to me that the simulations performed in this study with the chosen model does not completely allow to investigate some of the questions (for instance, what are the drivers of the increased atmospheric CO2 seasonal amplitude). The coupled climate – carbon cycle

model helps to understand the potential feedback between the land surface processes and the atmosphere and to investigate long term prediction; but the chosen model with its biases (i.e., the too low amplitude of the mean seasonal CO2 cycle) requires more caution when discussing the relative contribution of all potential drivers of the observed amplitude change (CO2, climate, agricultural changes, . . .).

It is not clear (at least to me) what the study brings in comparison to previous studies as I feel it does not focus enough on the "potential novelty" linked to i) the use of a coupled ESM as well as ii) the use of "regional atmospheric influence functions" to analyze the regional and temporal contribution of the potential drivers. Note that this last part is poorly valorized and not discussed in detail enough. I also find that on average the results are exposed but not analysed enough in terms of processes (GPP versus the different Respiration terms; contribution of different PFT; which are the key processes in the model that are responsible for the modeled trend and CO2 amplitude (water versus temperature limitations, . . .)). The limits of the model are also not discussed enough in terms of which scientific results are "robust" versus those that are likely not very uncertain (especially when discussing the time frame 2100 – 2300).

I thus recommend major revisions prior to consider that such work brings new information for the understanding and the prediction of the atmospheric CO2 seasonal amplitude changes.

Main comments

* Introduction:

- The authors provide a nice literature review of articles that have tried to explain the increase of atmospheric CO2 amplitude. However they lack the recent study by Hakihiko Ito et al., 2016 in Tellus " Decadal trends in the seasonal-cycle amplitude of terrestrial CO2 exchange resulting from the ensemble of terrestrial biosphere models Âż Note that such study is using an ensemble of process-based land surface models, including two version of CLM (CLM4 and CLM4VIC) which are probably close to CLM4CN

used in this study? Although such study was just published, it would be now crucial to include it in the literature review, given how comprehensive it is.

- Secondly and more importantly, we miss after such review what are the remaining critical uncertainties around the drivers of the seasonal CO2 increase? For instance Forkel et al. (2016) claimed that they could reproduce reasonably well the observed CO2 amplitude increase. What is thus missing or what is uncertain from their study? A critical analysis of the past literature in order to define the "niche" for this paper is missing. It would be good to have a set of more precise questions that the paper will target.

- Page 3, l13: The justification for the need of a full land-atmosphere-ocean coupled model is not provided, at least given the scientific questions that underlines the study? You need to justify why using the full ESM is beneficial and what can it bring compared to others studies (for instance, Ito et al. (2016) have used an ensemble of land surface models and similar experimental set up to separate the effect of potential drivers)? You could have envisaged forcing the CLM4CN model with climate predictions with a bias correction. What do you gain from your coupled approach ?

- It seems strange to me to emphasize the period 2100 – 2300 with a model that does not include Permafrost modeling and other critical processes linked to land management (no crop specific module, or no vegetation dynamic); while these may be more crucial in very long term simulations. You have at least to justify that the model is suitable to answer the question you pose.

- In general the introduction should propose a set of questions that follow from points that have not been treated by previous studies or based on the uncertainties that are still prevailing? And your approach (i.e. the use of CESM1) should be justified or at least explained with respect to the objectives.

* Model section:

What does CLM4CN do for natural vegetation shift. This will be crucial in the boreal zone with possible tree migration northward especially with such long time frame investigated (2300). Few word on this aspect would be beneficial.

* Experiement:

- The authors mention using "impose CO2" for the different experiment while in the result sections they say "The imposed emission scenario" (page 7, l10). The procedure became only clear to me when reading the note page 7, l15: "We note that the atmospheric CO2 moles fraction values were diagnostic only….". I thus think that the "experiment section" should describe more precisely what was done and differences between imposed CO2 and diagnostic CO2.

* Mapping atmospheric CO2 (section 2.3)

- Page 5, L 13: you should precise which patterns of the monthly CASA fluxes was used to prepare the pulse functions: GPP, NEP, NEE?

- Page 5, l25-28: There are potentially large differences between the CASA NEP spatial patterns and the CLM4CN ones so that it is not at all obvious that the "mapping approach with GEOS-Chem" will not be biased through differences in these spatial patterns. Discussion of Figure 2c brings a first insight but the authors should discuss more the impact of "surface pattern differences" and "transport differences" for the trend in the atmospheric CO2 amplitude rather than for the amplitude itself.

* Results

- Page 6, 30: It is not clear when you compare the 425 ppm simulated by CESM to the observed 391 ppm in 2010, over which period the drift occured (missing sink). This would need to be clarified so that we see more how much is he missing sink per year ?

- Page 7, L11: As I said above, you mention the "imposed emission scenario" but this is not detailed in the experiment section ?

- The change in surface temperature of 6K in 2100 and then 11K by 2300 makes me wonder about the prediction of the CO2 amplitude increase. With such large temperature change after 2100, neglecting permafrost melt and potentially large natural vegetation change in the artic may be severe limitation ? At least this should be discuss to gain confidence that the other effect accounted for are the primary ones ?

- More generally the fact that the model simulate only $\frac{1}{2}$ of the seasonal cycle atmospheric amplitude at high latitude is probably a strong limitation to study the "drivers of the amplitude increase". This should be discussed in more detail. Such a bias has probably large implications on the relative contribution of atmospheric CO2 increase, versus climate and land use change ?

- Page 8, L28: Why do you think that you still obtain a strong fertilization effect on the amplitude increase even given that CLM4CN has the lower fertilization effect of last CMIP5 models? Maybe you should explain a bit more which processes are contributing? Only the GPP increase? or other effects linked to autotrophic and heterotrophic respirations ?

- Climat change effect (section 3.3.2): I feel that not enough insight on the processes that lead climate change to impact the changes in atmospheric CO2 amplitude are given ? What is the role of the different respiration terms versus photosynthesis ? Do you see different contribution between grass and tree PFTs? What are the mechanism in CLM4CN that explain the contribution (sensitivity of the maximum photosynthetic uptake to temperature ?)

- LUC effect (section 3.3.3): It seems strange to mention that "the model is providing contrary results to previous studies" with the explanation that it does not properly treat cropland! At least we need a discussion to show that the current LUC effects are not completely wrong given such "model shortcut". The authors should detail why they think the other component of the LUC effect may be important and why "their message about LUC effect" is still a valuable one ?
- Page 9, l34: Precise over which period the growing season length increased by 1 month. 0verall the section 3.3.4 on the growing season length is not bringing much information. You could explain what contributes in the model to the change in growing season length (earlier starts or later end of the season). As for the tropic and the argument on the water use efficiency, you could provide more support by discussing how the soil moisture has evolved in the simulation with climate change.

* Discussion:

- Page 10, l22-23: You mention that the CESM has probably a too strong $CO_2$ fertilization effect. This is not intuitive as you previously mentioned that CLM4CN has the lowest fertilization effect from the CMIP5 suite of model and that it provides a too low mean amplitude and mean amplitude trend for high latitude. The reasoning and conclusion should be more detailed as it is not intuitive. You can have several compensating effects so that the fertilization in the model is not too strong. Also, what is potentially missing is a discussion of the fertilization effect in CLM4CN with respect for instance to "FACE" experiment to put in perspective the results and conclusion drawn for the 23 century.

- Page 10, l28: You mention that LUC reduced the amplitude of atmospheric $CO_2$ seasonal cycle, contrary to previous studies. You should indicate why the simulation with CESM provides new plausible information, given that you have mentioned that "treating crop as grassland" is a severe limitation (see my comment above). You have to provide some explanation on why you think the results with CESM provide a new perspective with respect to LUC. Basically what was the typical LUC that is contributing to such decrease, through which processes, . . . ?

- Page 11, last paragraph: Further discussion on the fact that "the results indicate that there is no high-temperature tipping point at which terrestrial productivity declines" would be valuable. Does this mean that the temperature dependence of the maximum photosynthesis peaks at high enough temperature threshold? or is it linked to the

nitrogen cycle ?

---

## Author Comment (AC1) · 8 Aug 2016

We thank the reviewer for her/his supportive review of our manuscript. Her/his suggestions will help us to strengthen the revised manuscript. In the response below, we address her/his concerns sequentially, with our responses indicated in "Arial" font, whereas her/his comments are indicated in "Times New Roman" font.

This study explores different contributors to the increase in the atmospheric $CO_2$ seasonal amplitude, as predicted by the CESM in simulations that span 1950—2300. I am generally supportive of this paper. Clearly an impressive effort went into it and it is well organized and written. However, I have some major concerns, listed in order of decreasing priority:

1) There are some steps in the methodology that need more detail and justification – some of them could/should really be stand-alone papers. These include a) The pulse-response method. b) The documentation of mid-latitude trends in observed $CO_2$ amplitude

We have added additional documentation, detailed below, to demonstrate the pulse-response approach we used to calculate atmospheric $CO_2$ mole fraction data. Because the focus of this paper is not on attributing drivers of the observed change in the amplitude, but rather exploring how seasonality changes in the future in a prognostic ESM, we prefer to minimize the discussion of observed mid-latitude trends.

2) The CESM does a poor job of reproducing the current $CO_2$ amplitude and the historical observed amplitude trends, which undermines confidence in the results presented here. Although I think the exercise is still worthwhile, some sort of well thought out rationale or statement is needed to explain why readers should believe or pay any heed to the future model results going out to 2300, e.g., are there certain results that are robust and insightful despite the model's poor present-day performance?

The exercise of predicting carbon-climate coupling in a fully prognostic model is still relatively new. Although CESM shows significant deficits in the simulated mean annual cycle and its trend, the model includes parameterizations for many of the processes that may be important in controlling its change with time, and we note that CESM qualitatively captures the northward increase in the NH atmospheric $CO_2$ seasonal amplitude as well as the increasing trend in the annual seasonal cycle amplitude. These suggest that the parameterizations included in CESM can be used to examine how the seasonal amplitude might evolve when subject to the radiative and fertilization effects of increased atmospheric $CO_2$ concentration, and also to identify deficiencies in current model parameterizations. While we do not expect that the simulation provides an accurate description of either carbon cycling or physical climate out to 2300, the results of the simulation do allow us to (1) understand the balance of major drivers, (2) identify deficiencies that may need to be addressed in future model development.

We will add the following statement to the discussion of the revised manuscript:

"Although CESM does not quantitatively reproduce the contemporary mean annual cycle amplitude or its trend over the last 50 years, parameterizations in the model qualitatively reproduce diagnostics such as the increase in both the mean annual cycle and its multi-decadal trend. Thus, we can use the model to understand partitioning of the long-term response to climate change or to fertilization, with an eye toward identifying areas for future model improvement."

Expanding on 1a) The pulse-response method. This could really be a stand- alone paper (see, e.g., Nevison, C.D., D.F. Baker, and K.R. Gurney, A methodology for estimating seasonal cycles of atmospheric $CO_2$ resulting from terrestrial net ecosystem exchange (NEE) fluxes using the Transcom T3L2 pulse-response functions, Geosci. Model Dev. Discuss., 5, 2789-2809, 2012,

www.geosci-model-dev- discuss.net/5/2789/2012/ doi:10.5194/gmdd-5-2789-2012, 2012.)

While I support the method and realize that it would be prohibitively expensive computationally to break down the contributions to $CO_2$ amplitude change from different regions and mechanisms without some sort of shortcut approach like the Pulse Response method, I think it needs more than a 1 paragraph explanation. For example:

i) Is there any IAV in the meteorology used to create the pulse fields? Also, what is the consequence of assuming those met fields will still apply in 2300?

We appreciate that the reviewer recognizes that the pulse-response method is a necessary computational shortcut to examine the regional contributions to atmospheric $CO_2$, and will provide more details about the method in the revised manuscript. The pulse response approach does contain interannual variability in the met fields, but as the reviewer identifies, there are substantial consequences in assuming those met fields will still apply in 2300. We note that in our manuscript Fig. 2c, mismatches grow from 2 ppm to 3 ppm in the mid- and high latitudes when the land $CO_2$ tracer in CESM (4-d) is sampled at the sites we use in Fig. 1 vs when NEE (which embodies all the land processes that influence the land $CO_2$ tracer) is convolved with the pulse-response function. This deficiency should be identified in the paper, and we plan to add the following text to our description of the pulse-response method:

"Although the CESM simulated the three-dimensional structure of atmospheric $CO_2$, we used a pulse-response transport operator to separate the imprints of $CO_2$ fluxes from different regions on the hemispheric $CO_2$ patterns. The transport operator was developed using the GEOS-Chem transport model (version 9.1.2, Nassar et al. (2010)). GEOS-Chem was configured as in Keppel-Aleks et al. (2013) on a $4^{\circ} \times 5^{\circ}$ horizontal grid with 47 vertical layers, and forced with meteorology fields from the 3–6-hourly Modern Era Retrospective-Analysis for Research and Applications (MERRA) reanalysis dataset (Rienecker et al., 2011). A tagged 1 Pg C month$^{-1}$ pulse was released for each of the 20 terrestrial source regions in Fig. 1 for each calendar month. Each 1 Pg C month$^{-1}$ pulse was distributed spatially according to monthly fluxes from the Carnegie-Ames-Stanford Approach (CASA) fluxes from Olsen and Randerson (2004). At a given location, the magnitude and phasing of the atmospheric $CO_2$ response of the pulse depends on the characteristics of atmospheric transport. For example, at Barrow (BRW) in Northern Alaska, a 1 Pg pulse released in Boreal North America has a large impact on atmospheric $CO_2$ (2 ppm, Fig. SA1a) during the first 1-3 months after a pulse is released. In contrast, when the pulse is released from temperate North America, there is a phase lag of 1 month (Fig. SA1b,c), and when the pulse is released from the Amazon, there is a delay in the peak response at BRW of 3 months (Fig. SA1d). The magnitude of the response is also smaller (e.g., 0.02 ppm for a 1 Pg pulse released in the Amazon vs 2 ppm for boreal North America; Fig. SA1), since the pulse has already diffused over much of the globe. We also note that seasonal patterns in atmospheric transport affect the imprint a pulse leaves on atmospheric $CO_2$ (Fig. SA2). For example, a 1 Pg pulse from the boreal region leaves a 2 ppm contribution on $CO_2$ at Barrow during the winter months, but more rigorous vertical mixing in the summer months reduces the imprint to 0.5 ppm. Following the twelve month period in which pulses were released, the signals were allowed to decay for 60 subsequent months, at which point $CO_2$ was well-mixed in the atmosphere (Fig. SA2a-d).

We then sampled GEOS-Chem at the locations of 41 NOAA cooperative $CO_2$ flask sample sites (Dlugokencky et al. (2013); Table 1, Fig. 1) for each month simulated. This resulted in a $CO_2$ transport operator matrix with dimensions $N_{reg.} \times N_{obs.} \times N_{mon.}$. We aggregated NEP fluxes from CLM4 to the spatial scale of the 20 source regions (Fig. 1), and used matrix multiplication to propagate these fluxes to atmospheric $CO_2$ space. We calculated the monthly mean $CO_2$ mole fraction at the observation sites by summing over the regional contributions to get a $CO_2$ response matrix with dimensions ($N_{obs}$ x $N_{mon}$).

We analyzed both the $CO_2$ fields from global fluxes and $CO_2$ patterns influenced only by larger regions

representing Arctic, boreal, temperate, subtropical, tropical, and Southern Hemisphere (SH) ecosystems. We calculated the $CO_2$ annual cycle amplitude values as the peak-to-trough differences in $CO_2$ summed over each component region (e.g., the $CO_2$ annual cycle amplitude at a given station from pulses emitted from the Arctic was calculated as the peak-to-trough difference in the sum of $CO_2$ from pulses emitted by the blue regions in Fig. 1). We note that our analysis focuses on surface observations of atmospheric $CO_2$, and does not include aircraft measurements.

The advantage of this method is that we can efficiently compute the regional contribution to changes in atmospheric $CO_2$; it would be prohibitively expensive to run a full atmospheric transport model for each of the regions separately for 350 years. To evaluate this method, we show a comparison in which we have generated $CO_2$ using NEE, since the land $CO_2$ tracer in the CAM4 is derived from NEE (despite that we use NEP for subsequent analyses). The magnitudes generally differed by less than 2 ppm due to different model boundary layer schemes and atmospheric transport (Fig. 2c). We note that the largest differences were during the last century of the simulation, which we hypothesize was due to shifts in atmospheric transport in response to the dramatic climate change in the CAM4. The fact that long-term trends in transport are not simulated by the pulse-response approach is one of the major sources of bias. By neglecting long-term trends in transport, we induce a bias into atmospheric $CO_2$ that increases with time (Fig. 2c). In a site-by-site comparison (Fig. SA3), the mismatch in time appears to be due to amplification of existing biases in the pulse-$CO_2$ compared to the full transport-$CO_2$. A second source of uncertainty is that the spatial distribution of fluxes within each region is different in CESM compared to CASA. We expect that this has a minimal impact based on results from Nevison et al. (2012), who showed that a similar pulse response code using different transport models did a reasonable job ($r^2$=0.8) of simulating the fossil fuel influence on $CO_2$ despite that $CO_2$ has a vastly different spatial configuration than do ecosystem fluxes.

We also assessed the validity of the assumption to model only the land contributions to trends in the mean annual cycle of $CO_2$ by calculating the $CO_2$ amplitudes in the CAM land and ocean tracers. We found that the contemporary peak-to-trough amplitude in the ocean tracer averaged across our high latitude stations was 2 ppm (in contrast to 10 ppm in the land tracer). Although both the land and ocean amplitudes grow with time, by 2300, the high latitude ocean tracer had an amplitude of 3 ppm, only 18% of the land amplitude for this time period. Ocean carbon uptake was found to change significantly in CESM through 2300 (Randerson et al., 2015), but based on these numbers, ocean $CO_2$ still had a smaller imprint on the atmospheric annual cycle."

ii) How are the 60-month decaying pulses combined to create a model atmospheric $CO_2$ cycle?

We have addressed the reviewer's question in the revised text, above, and created two additional figures (Fig. SA1 and Fig. SA2) to show this process graphically.

iii) In figure 2, the pulse-response amplitudes at midlatitudes are 3 ppm or more smaller than the fully prognostic tracer. This doesn't seem "broadly similar" and undermines confidence that this methodology can detect subtle trends, esp. in the midlatitudes.

To provide better validation for the reader to assess the bias induced by the pulse-response method, we have prepared Fig. SA3, which shows the mean seasonal cycle at a high-latitude (BRW), mid-latitude (SHM), subtropical (KEY), and tropical (MLO) NH site. These sites were selected since they have observational records dating to the 1980s (gray circles shown in Fig. 1). We plot both the CESM land $CO_2$ tracer and the pulse-response $CO_2$ for four periods for each site: 1990—1999, 2090—2099, 2190—2199, and 2290—2299.

The site-by-site comparison shows that (1) the biggest mismatches in 2300 between the full-transport and the pulse-response $CO_2$ owe to persistent biases that exist for the present, e.g., the high January bias at Barrow (BRW) and the one-month phase shift in the summer minimum at KEY. This would suggest that changes in transport patterns due to climate change induce a smaller mismatch than present-day biases

in the method. (2) the method is able to capture fairly subtle variations in the mean annual cycle, such as the "W" shape that the mean annual cycle at SHM develops over time.

iv) The GMD Discussions paper above was never accepted for final publication, due to reviewers who thought adjoint methods were superior. While the current method is superior in that it divides land into a larger number of regions (20 v. 11), the GMDD paper on the other hand was applying the method to estimate mean seasonal cycles, which are easier to get right than the more subtle trends in amplitude over time examined here.

We agree with the reviewer that not only are the improved resolution of land areas an advantage of our pulse-response code over the Transcom regions, but also the fact that these land regions were determined based on similarity in annual mean NPP and its seasonality. We agree that comparison of mean annual $CO_2$ cycle, rather than its trend, places a lower burden on the code. For this application, however, the mean annual $CO_2$ amplitude changes by up to 10.6 ppm by 2300. Thus, the relative error, assuming a 3 ppm difference, is still only ~28% of the total trend.

Expanding on 1b) I'm not sure there is any evidence that $CO_2$ seasonal amplitude is increasing at midlatitude sites such as NWR or UUM, KZM/D. In fact, if anything, they may be decreasing – possibly due to drought effects. The most robust effects are seen at BRW, with the amplitude increase at MLO less than half that of BRW. I don't think Zeng et al. (2014) is an adequate reference to prove that midlatitude $CO_2$ amplitude is increasing, since they don't actually show this.

We thank the reviewer for this comment. One of the reasons we aggregate the sites depicted in Fig. 1 into high-, mid-, subtropical, and tropical latitude belts is to minimize local effects that may be present at the sites we have chosen and to instead focus on a more large-scale pattern of variation. A challenge to this approach is that there are relatively few ESRL sites with records dating to the 1980s or earlier. We will include this rationale in our methods discussion by including the text:

"In our analysis, we aggregate the sites into high-, mid-, subtropical, and tropical latitude belts to minimize local effects at individual sites and instead to focus on large-scale trends owing to broad patterns of climate change."

We will also thoroughly check our referencing in the revised paper and be sure to reference observationally based papers such as Randerson et al. (1997) and Graven et al. (2013) rather than modeling-derived studies.

Minor comments: p.1, L8, The term "changing atmospheric composition" to encompass $CO_2$ fertilization and N deposition is confusing. These two don't really belong in the same category, in my opinion, since the N deposition is relevant mainly after it deposits on the soil, i.e., the authors are not looking at some sort of physiological response of plants to increased atmosphere NOx or NH3 concentration.

We agree with the reviewer that $CO_2$ fertilization and N-deposition represent two distinct forcings on ecosystem carbon exchange. Unfortunately, the simulations which were conducted as part of the CESM Biogeochemistry Working Group did not separate these two changes, thus we cannot resolve the specific forcing from the model output available since there is no analogous CESM ECP simulation that excludes N-deposition and the $CO_2$ radiative effect. In each of the ECP simulations, reactive nitrogen deposition was kept constant at 2100 values (Randerson et al., 2015), so future trends in the mean annual cycle amplification were due to nitrogen deposition levels at 2100 interacting with trends in $CO_2$.

Devaraju et al. (2016) did perform experiments looking at the individual and combined effects of $CO_2$

fertilization, N-deposition, climate change, and LUC on historical NPP trends using the CESM1(BGC). They found that $CO_2$ fertilization and N-deposition contributed 2.3 and 2 PgC $yr^{-1}$ to the 4 PgC $yr^{-1}$ historical increase in global terrestrial NPP. Given the conditions of the experiments and the fact that $CO_2$ fertilization and N-deposition contribute similarly to global and historical NPP trends in the CESM, we present our results based on the combined effects of $CO_2$ fertilization and N-deposition.

We will reference this paper in the text, and include the following statement:

"Results from Devaraju et al. (2016) suggest that global NPP is influenced equably by $CO_2$ fertilization and nitrogen deposition over the historical period in CESM, so trends in the mean annual cycle amplitude were likely influenced by this enhanced NPP. In these simulations, nitrogen deposition was held fixed after 2100, so trends in the amplitude were influenced by anthropogenic nitrogen deposition but not forced by transient deposition."

We will also use clearer language to describe that these two effects are included in the simulations by replacing "changing atmospheric composition" with "$CO_2$ fertilization and N-deposition" throughout the revised manuscript.

p.1, L12 is confusing as written – in one case we have the end time (2300) and in the other we have the start time (after 2100). Please rewrite to clarify start and end times for both effects

We will revise the sentence to read "$CO_2$ fertilization and N-deposition in NH boreal and temperate ecosystems were the largest contributors to mean annual cycle amplification over the midlatitudes for the duration of the simulation (1950—2300) and for the Arctic from 2100—2300."

p.1, L15 "rather than the strength of the terrestrial carbon sink" please explain more clearly what is meant here.

We will clarify this sentence to read "Greater terrestrial productivity during the growing season was the largest contributor to the annual cycle amplification throughout the Northern Hemisphere."

p.1, L17, suggest replacing "is not predicated on" with "does not necessarily imply" p.1, L20 I think it's more accurate to say "at some NH sites" rather than "over the NH" (see my comments above about midlatitude trends).

We will change the sentence to read "Prior to 2100, $CO_2$ annual cycle amplification occurred in conjunction with an increase in the NH land carbon sink, but these trends decoupled after 2100, underscoring that an increasing atmospheric $CO_2$ annual cycle amplitude does not necessarily imply a strengthened terrestrial carbon sink."

p.2, L31 missing AND between citations.

We will add the "and" between McDonald et al. (2004), and Barichivich et al. (2013).

p.2, L35 suggest saying, "Model evidence suggests that the combined effects . . ." and delete "in simulations."

We will revise the sentence to read, "Model evidence suggests that the combined effects of climate change and shifts in vegetation cover can also enhance GPP."

P2., L20 and p.3, L17 again I find the catch-all term "changes in atmospheric composition" confusing.

We will refer to "changing atmospheric composition" as "$CO_2$ fertilization and N-deposition".

p. 6, L30. It seems like a stretch to call 425 ppm and 391 ppm "roughly equivalent"

We agree with the reviewer and will modify the text in the revised manuscript to state "We note that the drivers of the amplitude increase during 1985—2013 were simulated to different levels of fidelity: the NH atmospheric temperature increase over land was roughly equivalent (1.02 K vs 0.95 K in the NCEP-NCAR reanalysis [Kalnay et al., 1996]), but the NH atmospheric $CO_2$ mole fraction in CESM was too high (425 ppm vs 391 ppm). Previous analysis of the CESM shows that the high $CO_2$ bias is attributable to persistent weak uptake in both land and ocean (Keppel-Aleks et al., 2013; Long et al., 2013)."

p.7, L5 Please provide a reference for the observed mid-latitude trend of 0.04 ppm $yr^{-1}$.

We thank the reviewer for calling this detail to our attention. We calculated the 0.04 ppm $yr^{-1}$ midlatitude trend from the 1985—2013 monthly observations at Shemya Island, Alaska (SHM), which we selected to represent the midlatitudes (40°N—60°N; Table 1) based on its sufficiently long period of record.

We will revise the text to clarify "Both the modeled and observed trends in the $CO_2$ annual cycle amplitude were calculated from individual sites whose records date to 1985 (gray circles in Fig. 1). The modeled trend in the $CO_2$ annual cycle amplitude..."

P8, L19 Please explain further. Why is this consistent with effects being proportional to GPP?

Regional GPP is smaller in the Arctic than in the other regions we analyze in the paper, thus fertilization acts as a knob on a smaller gross flux term. We will revise the statement to read, "$CO_2$ fertilization and N-deposition effects were smallest in the Arctic, the region with the smallest GPP for the contemporary period. In the CESM, the impact of $CO_2$ fertilization on the amplitude trend roughly scales with to the magnitude of overall GPP, consistent with hypotheses from Tans et al., (1990) and Schimel et al (2015) that the fertilization effect on the land carbon sink is proportional to productivity."

P9, L12, to avoid confusion, would suggest splitting into 2 sentences: "..simulation. These latter influences added 4.7 ppm . . ."

We will split these two sentences in the revised manuscript.

P9, L27 The Zeng et al reference, in my reading, does not actually demonstrate that the spatial distribution of where atmospheric $CO_2$ amplitude increases are seen (mainly at high latitudes) are consistent with agriculture, which is large at mid-latitudes.

We agree with the reviewer that the Zeng et al. (2014) reference does not explicitly calculate the impact of midlatitude agricultural fluxes on the high latitude mean annual cycle amplitude, where the trend is largest. However, our results (Fig. 7) show that temperate ecosystems leave a large imprint on the mean annual cycle amplitude at high latitudes. Thus, if crops were included in the CESM, the model would show the imprint at high latitudes. We therefore prefer to leave the statement unchanged.

P10, L23, "perhaps indicating . . ." Please explain further.

Based on comments from both reviewers, we have decided to revise the text to remove this statement. Instead, we will include a statement that this finding demonstrates the importance of considering latitudinally resolved $CO_2$ in models for diagnosing compensating errors. As described in the response to Reviewer 2, one difference between our paper and other papers on the mean annual cycle is that we explicitly consider how fluxes propagate to atmospheric $CO_2$ rather than simply aggregating hemispheric fluxes.

We will revise the text to read: "This result underscores the importance of considering meridionally resolved atmospheric $CO_2$ data that explicitly considers the role of transport, since a Northern Hemisphere average masks incorrect spatial patterns in the CESM."

Additional References

Devaraju, N., Bala, G., Caldeira, K., and Nemani, R.: A Model Based Investigation of the Relative Importance of $CO_2$-Fertilization, Climate Warming, Nitrogen Deposition and Land Use Change on the Global Terrestrial Carbon Uptake in the Historical Period, Clim. Dynam., 47, 173—190, doi:10.1007/s00382-015-2830-8, 2016.

Nevison, C. D., Baker, D. F., and Gurney, K. R.: A methodology for estimating seasonal cycles of atmospheric CO2 resulting from terrestrial net ecosystem exchange (NEE) fluxes using the Transcom T3L2 pulse-response functions, Geosci. Model Dev. Discuss., 5, 2789–2809, doi:10.5194/gmdd-5-2789-2012, 2012.

Tans, P. P., Fung, I. Y., and Takahashi, T: Observation constraints on the global atmospheric $CO_2$ budget, Science, 247, 1431–1438, 1990.

Olsen, S. C., and J. T. Randerson: Differences between surface and column atmospheric $CO_2$ and implications for carbon cycle research, J. Geophys. Res., 109, D02301, doi:10.1029/2003JD003968, 2004.

[Figure]

**Fig. SA1**: The imprints of 1 Pg pulses emitted in 12 successive months (x-axis) from (a) NBNA, (b) ETNA, (c) WTNA, and (d) AMZN on the atmosphere sampled at BRW over a 60-month period (y-axis).

[Figure]

**Fig. SA2**: The imprints of 1 Pg pulses emitted from (a) NBNA, (b) ETNA, (c) WTNA, and (d) AMZN in each month (contours) on the atmosphere sampled at BRW.

[Figure]

**Fig. SA3**: Mean annual cycles of atmospheric $CO_2$ derived from (blue curves) NEE run through the pulse response function and (black curves) the CESM land $CO_2$ tracer for (a—d) BRW, (e—h) THD, (i—l) KEY, and (m—p) MLO in 1990—1999, 2090—2099, 2190—2199, and 2290—2299.

---

## Author Comment (AC2) · 8 Aug 2016

We thank the reviewer for her/his constructive review of our manuscript. We found the comments very helpful, as they revealed that in the revised manuscript we must revise our language to show the insights gained in understanding how the mean annual cycle of $CO_2$ changes in response to climate and environmental drivers in a fully coupled ESM. The review underscored that the rationale for our study was not made clear, and we will remedy this in the revised manuscript. Here, we respond point-by-point to the reviewer's comments (Times New Roman font) with our rationale and proposed modifications to the revised text (Arial font).

The paper describes an analysis of potential drivers of multi-century trends in the seasonal cycle amplitude of the atmospheric $CO_2$ concentration with a Prognostic Earth System Model. The study follows from the paper of Graven et al. (2013) that analyzed in detail the large increase of atmospheric $CO_2$ seasonal cycle amplitude at high northern latitudes over the past 60 years; In a series of studies trying to disentangle the drivers of the observed increase in atmospheric $CO_2$ seasonal amplitude, this paper propose a first attempt with a prognostic coupled carbon-climate cycle model and an investigation of the amplitude changes up to the horizon 2300.

While our study follows from several papers [e.g., Randerson et al. (1997) and Graven et al. (2013)] that showed that the mean annual cycle of $CO_2$ at high northern latitudes has increased steadily since measurements began in 1958, the goal of our paper is less to attribute drivers of the observed increase and more to test the abilities of a prognostic ESM to simulate the increase and to explore whether the ESM predicts nonlinearities or tipping points in the long-term increasing trend as climate in the model continues to evolve past the present-day. The climate and biogeochemical communities have invested tremendous time into the development of fully-coupled, mechanistic models, and rarely has a multi-decadal phenomenon such as the long-term $CO_2$ amplitude increase been observed in nature and therefore provided an opportunity to test a multi-decadal model in a fully-coupled, prognostic model.

The paper is clearly written and relatively easy to follow. However, it seems to me that the simulations performed in this study with the chosen model does not completely allow to investigate some of the questions (for instance, what are the drivers of the increased atmospheric $CO_2$ seasonal amplitude). The coupled climate–carbon cycle model helps to understand the potential feedback between the land surface processes and the atmosphere and to investigate long term prediction; but the chosen model with its biases (i.e., the too low amplitude of the mean seasonal $CO_2$ cycle) requires more caution when discussing the relative contribution of all potential drivers of the observed amplitude change ($CO_2$, climate, agricultural changes, . . .).

We agree with the reviewer that more discussion of biases in the version of CESM run for this study requires additional attention in the reviewed manuscript. We also recognize that we need to reframe discussion away from "what drove the observed amplitude" and more toward "what nonlinearities are present in a prognostic ESM that influences its ability to simulate multi-decadal through multi-century trends in coupled climate-carbon cycling?".

It is not clear (at least to me) what the study brings in comparison to previous studies as I feel it does not focus enough on the "potential novelty" linked to i) the use of a coupled ESM...

i) We thank the reviewer for her/his helpful comments here that prompt us to recognize the need for us to provide better framing for our study's motivation and results. The use of a coupled ESM is crucial for our major goal, which is to explore whether there are changes in drivers of the mean annual cycle amplitude in a future climate change. We will add the following text to the introduction: "The use of a coupled model allows us to simulate the co-evolution of physical climate and biogeochemistry using a self-consistent framework. This is crucial since carbon fluxes are inherently linked to the physical climate; for example, a

change in GPP will be associated with changes in evapotranspiration, which feeds back on metrics such as humidity, cloud cover, and precipitation. Moreover, in a fully prognostic model, both climate and carbon cycle diagnostics are free to evolve rather than being tied to input data sets that reflect the contemporary climate."

... as well as ii) the use of "regional atmospheric influence functions" to analyze the regional and temporal contribution of the potential drivers. Note that this last part is poorly valorized and not discussed in detail enough.

ii) In response to this comment, and some comments from Reviewer 1, we plan to add additional figures to demonstrate the pulse response methodology and validation against the full-transport land $CO_2$ field simulated by CESM. We will also add the text included in our response to Reviewer 1 to better explain and validate the method.

I also find that on average the results are exposed but not analysed enough in terms of processes (GPP versus the different respiration terms; contribution of different PFT; which are the key processes in the model that are responsible for the modeled trend and $CO_2$ amplitude (water versus temperature limitations, . . .)). The limits of the model are also not discussed enough in terms of which scientific results are "robust" versus those that are likely not very uncertain (especially when discussing the time frame 2100–2300).

We address these drivers in our responses to the reviewer's individual comments below.

I thus recommend major revisions prior to consider that such work brings new information for the understanding and the prediction of the atmospheric $CO_2$ seasonal amplitude changes.

Main comments

* Introduction:

- The authors provide a nice literature review of articles that have tried to explain the increase of atmospheric $CO_2$ amplitude. However, they lack the recent study by Hakihiko Ito et al., 2016 in Tellus " Decadal trends in the seasonal-cycle amplitude of terrestrial $CO_2$ exchange resulting from the ensemble of terrestrial biosphere models" Note that such study is using an ensemble of process-based land surface models, including two versions of CLM (CLM4 and CLM4VIC) which are probably close to CLM4CN used in this study? Although such study was just published, it would be now crucial to include it in the literature review, given how comprehensive it is.

We thank the reviewer for bringing this article to our attention, and will include discussion of this article in the revised manuscript. We note that this article was published after our manuscript was published in *Biogeosciences Discussions*, so we did not have the opportunity to include it in our initial submission. Likewise, another relevant paper was published in *Biogeosciences Discussions* a few days after our initial submission. These papers are valuable in that both us multi-model ensembles (MsTMIP and TRENDY, respectively) to consider changes in the mean annual cycle of land-atmosphere carbon fluxes.

An important difference between these papers and our manuscript is that these papers focus on the seasonal cycle of fluxes, rather than propagating those fluxes to atmospheric $CO_2$ concentration, which we focus on in our paper. The propagation of fluxes to atmospheric $CO_2$, even using a simple method such as the pulse response code that we use, is important since atmospheric transport plays a major role in the spatial gradient in atmospheric mean annual cycle trends. We also note that in terms of

understanding future observations, we cannot directly observe GPP (although promising remote sensing tools such as chlorophyll fluorescence are being developed) or ecosystem respiration at large spatial scales. Thus, using a model such as CESM to develop hypotheses about how individual process might change the quantity that *is* directly observable (atmospheric $CO_2$) is a valuable exercise, in our opinion.

We will revise the introduction to include discussion of these manuscripts and to differentiate our approach from these papers. The following text will be inserted before the paragraph on p3, L10: "Several recent papers have considered how the amplitude of NH net carbon exchange has changed over the historical period. Ito et al. (2016) analyze MsTMIP terrestrial ecosystem models to determine how atmospheric $CO_2$, climate change, and land use affect the NH flux amplitude for the historical period, and Zhao et al. (2016) analyze the net terrestrial flux to the atmosphere in TRENDY models. Both of these studies find that $CO_2$ fertilization is the strongest driver of increasing ecosystem productivity and thus the amplitude of the net carbon exchange in the NH. The results from these ensemble-based analyses provide a useful basis for comparison for our analysis of a single, fully coupled ESM. A significant difference between the approach used by these papers and our study is that they consider the net flux amplitude, whereas we propagate fluxes using an atmospheric transport operator to determine the influence on latitudinally resolved atmospheric $CO_2$ fields. We hypothesize that fingerprints of climate change or $CO_2$ fertilization may be evident in different latitude bands in the CESM output."

This statement will be followed by the following text, to be included in the discussion section of the revised manuscript:

"CESM simulations show that the major drivers of the mean annual cycle amplification leave differential imprints on atmospheric $CO_2$ in different latitude bands. For example, $CO_2$ fertilization leaves the largest imprint in both absolute and relative terms on midlatitude $CO_2$, whereas climate change may amplify high latitude $CO_2$ while having a near-neutral impact on $CO_2$ annual cycle amplitudes south of 60°N (Fig. 10). These fingerprints may be useful for developing hypotheses regarding observed trends and determining future observational strategies to monitor carbon-climate feedbacks."

- Secondly and more importantly, we miss after such review what are the remaining critical uncertainties around the drivers of the seasonal $CO_2$ increase? For instance, Forkel et al. (2016) claimed that they could reproduce reasonably well the observed $CO_2$ amplitude increase. What is thus missing or what is uncertain from their study? A critical analysis of the past literature in order to define the "niche" for this paper is missing. It would be good to have a set of more precise questions that the paper will target.

We thank the reviewer for this constructive comment. An implicit premise of our study was that the drivers of changes in the mean annual cycle between 1958 and 2013 need not continue to drive changes in the mean annual cycle of $CO_2$ into the future. Fertilization impacts could saturate, while further increases in temperature or related changes in drought conditions may actually reverse trends in seasonal productivity. We recognize that we need to make this premise more explicit in the revised manuscript. The Forkel et al. (2016), Ito et al. (2016), and Zhao et al. (2016) studies all focus on explaining only the historical trend, not future projections. We choose to study this topic in a single climate model so that we can in more detail analyze the regional contributions by driver to future changes in the mean annual cycle amplitude.

In the Ito et al. (2016) paper, a fair amount of attention is given to the idea that the mean annual cycle strength correlates with the net terrestrial sink strength. Because the simulations were run to 2300, we are able to determine the time period in CESM where this statement is no longer true. In the extended concentration pathway simulations, the mean Northern Hemisphere $CO_2$ amplitude is correlated with increased Northern Hemisphere carbon uptake (using NEP and neglecting land use change, disturbance, and harvest fluxes) through ~2150, at which point NEP shows significant declines in the Northern Hemisphere while there are only small changes to the amplitude (Fig. SB1).

[Figure]

**Fig. SB1**: FullyCoupled atmospheric $CO_2$ annual cycle amplitudes ($A^{FC}$) versus NEP averaged over the NH and shaded according to simulation year. Negative NEP values indicates net carbon uptake by the land surface.

Based on the reviewer's comment, we plan to add a section on " Uncertainties and future model needs" to the discussion section to the paper, in which we explicitly discuss how lack of permafrost parameterizations, vegetation successional patterns, active human management, etc. affect the simulation results. It is exciting that CESM2, in preparation for the CMIP6 experiments, has much improved representation on frozen soil carbon and temperature interactions (Koven et al., submitted) as well as land management representation (P. Lawrence et al., BGCWG February Meeting presentation). Moreover, a version of CLM-ED will be released this fall. These new developments present opportunities for follow-on studies to explore the impact of these "missing" interactions. However, we feel that these comparisons are outside the scope of the current paper and are best reserved for a future study. Our paper, instead, provides a baseline analysis of the CESM1.

We also thank the reviewer for the suggestion of explicitly including questions that the paper will address. We will include the following questions at the close of the "Introduction" section of the revised manuscript:

"The questions guiding our analysis of CESM extended concentration pathway simulations are as follows:

1. Does the relative importance of drivers of the $CO_2$ amplitude trend change after 2100? For example, do we see evidence of saturation of the $CO_2$ fertilization effect or evidence of a climatic tipping point after which the $CO_2$ amplitude declines?

2. Do the regional contributions to $CO_2$ mean annual cycle trends change in response to large changes in climate?

3. Does the $CO_2$ annual cycle amplitude scale with the hemispheric carbon sink from NEP as climate and atmospheric conditions evolve in the future?"

- Page 3, l13: The justification for the need of a full land-atmosphere-ocean coupled model is not provided, at least given the scientific questions that underlines the study? You need to justify why using the full ESM is beneficial and what can it bring compared to others studies (for instance, Ito et al. (2016) have used an ensemble of land surface models and similar experimental set up to separate the effect of potential drivers)? You could have envisaged forcing the CLM4CN model with climate predictions with a bias correction. What do you gain from your coupled approach?

Since the goal of the paper is to explore future trends the use of a prognostic climate model is crucial since we do not have some bias-corrected estimate for climate change. Given the scientific questions we have added to the paper, and our response (above) for why CESM is a good tool for this analysis, we think that our approach has now justified in the manuscript text.

- It seems strange to me to emphasize the period 2100–2300 with a model that does not include Permafrost modeling and other critical processes linked to land management (no crop specific module, or no vegetation dynamic); while these may be more crucial in very long term simulations. You have at least to justify that the model is suitable to answer the question you pose.

We agree with the reviewer that there are limitations to the CESM configuration for the science questions we address in our paper, including the lack of permafrost modeling and land management. We note that ESM development is a slow and steady process, and that there is value to fully exploring processes in CESM1–the first fully coupled version of this model. Moreover, comparisons among different model versions are crucial, so careful analysis of CESM1 will provide better insights and science questions for subsequent analysis of CESM2.

We will add the following text to the introduction P3, paragraph ending on line 22: "The CESM provides a unique platform for exploring these questions in that it is one of the few prognostic ESMs to include coupled carbon-nitrogen biogeochemistry and diagnostic atmospheric $CO_2$ variability."

- In general the introduction should propose a set of questions that follow from points that have not been treated by previous studies or based on the uncertainties that are still prevailing? And your approach (i.e. the use of CESM1) should be justified or at least explained with respect to the objectives.

Per the reviewer's suggestion, in the revised paper we plan to introduce the following questions:

1. Does the relative importance of drivers of the $CO_2$ amplitude trend change after 2100? For example, do we see evidence of saturation of the $CO_2$ fertilization effect or evidence of a climatic tipping point after which the $CO_2$ amplitude declines?

2. Do the regional contributions to $CO_2$ mean annual cycle trends change in response to large changes in climate?

3. Does the $CO_2$ annual cycle amplitude scale with the hemispheric carbon sink from NEP as climate and atmospheric conditions evolve in the future?"

* Model section:

What does CLM4CN do for natural vegetation shift. This will be crucial in the boreal zone with possible tree migration northward especially with such long time frame investigated (2300). Few word on this aspect would be beneficial.

CLM4CN does not include dynamic vegetation. We will include the following text in the model description (Section 2.1): "These simulations were run without dynamic vegetation, which potentially also damps feedbacks that could contribute to changes in the $CO_2$ annual cycle through 2300."

We will also add text to the "Uncertainties and future model needs" section that will be added to the discussion: "The lack of dynamic vegetation in this version of CESM contributes some uncertainty to these results. Tree cover is expected to expand further northward with climate change (e.g., Lloyd et al.,

2005), which may contribute to the long-term increase in NEP flux amplitude within high latitude ecosystems. In contrast, drying at lower latitudes may lead to replacement of trees with grasses and subsequent decreases in NEP amplitude. Thus, the balance of these processes on the overall flux amplitude and spatial variability in the atmospheric $CO_2$ trend is uncertain. An ecosystem demography version (CLM-ED) is currently being developed that would permit successional patterns in response to environmental change. We consider the documentation of trends in the static-vegetation configuration presented in this manuscript to be a crucial first step toward eventually determining the sensitivity of land-atmosphere biogeochemical couplings in more sophisticated, future configurations of the CESM model."

* Experiment:

- The authors mention using "impose $CO_2$" for the different experiment while in the result sections they say "The imposed emission scenario" (page 7, l10). The procedure became only clear to me when reading the note page 7, l15: "We note that the atmospheric $CO_2$ mole fraction values were diagnostic only. ...". I thus think that the "experiment section" should describe more precisely what was done and differences between imposed $CO_2$ and diagnostic $CO_2$.

We will include this description in Section 2.2 "Experiments". The revised text will read: "The mole fraction of $CO_2$ in the atmosphere is prescribed according to the RCP8.5 and ECP8.5 scenario described by Meinshausen et al. (2011), and it is this value that controls radiative forcing as well as $CO_2$ fertilization. However, the CESM retains a separate, spatially-varying $CO_2$ tracer that is a diagnostic passive tracer of land, ocean, and fossil fuel carbon fluxes; the additional carbon exported from the surface to the atmosphere does not exert any forcing on the climate."

- Page 5, L 13: you should precise which patterns of the monthly CASA fluxes was used to prepare the pulse functions: GPP, NEP, NEE?

We will revise section 2.4 to state we used monthly mean NEP from the CESM to derive atmospheric $CO_2$ from the pulse response function.

- Page 5, L25—28: There are potentially large differences between the CASA NEP spatial patterns and the CLM4CN ones so that it is not at all obvious that the "mapping approach with GEOS-Chem" will not be biased through differences in these spatial patterns. Discussion of Figure 2c brings a first insight but the authors should discuss more the impact of "surface pattern differences" and "transport differences" for the trend in the atmospheric $CO_2$ amplitude rather than for the amplitude itself.

We have included revised text and figures in the response to Reviewer 1 to address these points. We anticipate that surface pattern differences are a minor source of disagreement since Nevison et al. (2012, GMDD) tested a similar pulse-response framework for fossil fuel emissions. The fossil emissions were distributed according to NEE, which represents a gross mismatch, but still had an $r^2$ value of 0.8 compared to a full transport simulation. Mismatches between CESM and CASA terrestrial fluxes are likely much smaller, although we have mentioned this factor as an additional source of error in the revised text.

* Results

- Page 6, 30: It is not clear when you compare the 425 ppm simulated by CESM to the observed 391 ppm in 2010, over which period the drift occurred (missing sink). This would need to be clarified so that we see more how much is he missing sink per year?

Previous results have shown that the missing sink for atmospheric $CO_2$ in CESM is attributable to weak

uptake in both the land and the ocean, and that this sink is relatively smooth with time.

We will revise the first paragraph of section 3.1 to conclude with "We note that the drivers of the amplitude increase during 1985—2013 were simulated to different levels of fidelity: the NH atmospheric temperature increase over land was roughly equivalent (1.02 K vs 0.95 K in the NCEP-NCAR reanalysis (Kalnay et al., 1996)), but the NH atmospheric $CO_2$ mole fraction in CESM was too high (425 ppm vs 391 ppm derived from observations in 2010). Previous analysis of CESM shows that the high $CO_2$ bias is attributable to persistent weak uptake in both land and ocean (Keppel-Aleks et al., 2013; Long et al., 2013)."

- Page 7, L11: As I said above, you mention the "imposed emission scenario" but this is not detailed in the experiment section?

We will clarify in the experiment section the details of the imposed emission scenario as described in our previous response.

- The change in surface temperature of 6 K in 2100 and then 11 K by 2300 makes me wonder about the prediction of the $CO_2$ amplitude increase. With such large temperature change after 2100, neglecting permafrost melt and potentially large natural vegetation change in the artic may be severe limitation? At least this should be discuss to gain confidence that the other effect accounted for are the primary ones

We agree with the reviewer that more discussion of this limitation, beyond noting the absence of permafrost dynamics in Section 2.1, should be addressed in our paper, and will add the following discussion to the "Uncertainties and future model needs" section of the revised manuscript: "The lack of permafrost dynamics likely has a large impact on $CO_2$ annual cycle trends, especially later in the simulation when global mean temperature has increased by over 10 K in the fully coupled simulation. Ongoing model development in CESM includes improved representation of permafrost carbon (Koven et al., 2015), and thus future model configurations will provide an improved tool for investigating a process that may provide one of the tipping points we identify in our key science questions."

- More generally the fact that the model simulate only of the seasonal cycle atmospheric amplitude at high latitude is probably a strong limitation to study the "drivers of the amplitude increase". This should be discussed in more detail. Such a bias has probably large implications on the relative contribution of atmospheric $CO_2$ increase, versus climate and land use change?

Many CMIP5 models exhibit the same bias (e.g., Zhao et al., 2016) show that the mean TRENDY model shows a 40% deficit in the annual mean). Since we intend the study to be an examination of what a fully prognostic model can tell us about trends, tipping points, and our current ability to simulate these interactions, we feel there is still value in quantifying drivers of trends within CESM1.

We will add the following text to Section 3.1, Line 27: "Although the CESM simulates low mean annual cycle amplitude throughout the NH, we note that many land models have a low bias in their simulated fluxes. For example, TRENDY land models show a 40% deficit in the magnitude of the seasonal cycle (Zhao et al., 2016)".

- Page 8, L28: Why do you think that you still obtain a strong fertilization effect on the amplitude increase even given that CLM4CN has the lower fertilization effect of last CMIP5 models? Maybe you should explain a bit more which processes are contributing? Only the GPP increase? or other effects linked to autotrophic and heterotrophic respirations?

We will add the following text to clarify that GPP is the main driver of the trends:

"Enhanced GPP seasonality appears to drive to the amplification of the atmospheric $CO_2$ seasonal cycle over northern temperate and boreal regions during 1950—2300. In midlatitude temperate regions where $CO_2$ fertilization drives the $CO_2$ seasonal cycle amplification, the seasonal amplitudes of GPP, HR and AR increase from 1950 to 2250, but the magnitudes of and increases in GPP seasonal amplitude are larger than those of HR, AR, and NEP in the FullyCoupled, NoRad, and NoLUC simulations. For example, in eastern temperate North America (ETNA), FullyCoupled GPP seasonal amplitudes increase from 6.8 PgC in 1950 to 11 PgC in 2250, while HR amplitudes increase from 0.85 PgC to 1 PgC, and AR amplitudes increase from 4 PgC to 7.6 PgC. The absolute increases in the seasonal amplitudes of GPP and total respiration (AR+HR) are, respectively, 2.5 and 2.4 times larger than the increase in the NBNA NEP amplitude during this period. Moreover, we find that GPP in high latitude regions, where climate change is the dominant contributor to amplification of net fluxes, is highly correlated with temperature. In the pulse regions that comprise our broader Arctic and boreal zones, GPP continues to increase with temperature until surface air temperatures surpass ~300 K."

- Climate change effect (section 3.3.2): I feel that not enough insight on the processes that lead climate change to impact the changes in atmospheric $CO_2$ amplitude are given? What is the role of the different respiration terms versus photosynthesis? Do you see different contribution between grass and tree PFTs? What are the mechanisms in CLM4CN that explain the contribution (sensitivity of the maximum photosynthetic uptake to temperature)?

We address the reviewer's comment in the response/text above.

In addition, we will include an analysis of the changes in PFT cover that contribute to the reduction FullyCoupled atmospheric $CO_2$ seasonal amplitudes from land-use change (LUC). As stated in the Methods section 2.2, PFT fractions vary on an annual basis from 1850—2100, then are held at 2100 values through 2300 in the FullyCoupled simulation. In the NoLUC simulation, PFT fractions are held at 1850 values. Crops (treated as unmanaged grass), needleleaf evergreen trees, and grass PFTs cover most of the NH boreal and temperate vegetated land. Between 1850 and 2100, boreal, temperate, and subtropical crop cover increase, while grass and needleaf tree cover decrease. Therefore, the decrease in the NH atmospheric $CO_2$ seasonal amplitude in response to temperate and boreal LUC reflects the fact that needleleaf evergreen tree and grass cover in these regions is lower in FullyCoupled than in NoLUC, resulting in lower GPP and smaller NPP seasonal amplitudes.

- LUC effect (section 3.3.3): It seems strange to mention that "the model is providing contrary results to previous studies" with the explanation that it does not properly treat cropland! At least we need a discussion to show that the current LUC effects are not completely wrong given such "model shortcut". The authors should detail why they think the other component of the LUC effect may be important and why "their message about LUC effect" is still a valuable one?

We agree that prescribing LUC and treating crops as unmanaged grass may produce unrealistic responses in atmospheric $CO_2$ seasonality. The contrast between our results, which show that LUC reduces NH atmospheric $CO_2$ seasonality and other studies, such as Zhao et al. (2016), showing that LUC increases atmospheric $CO_2$ seasonality indicate that more sophisticated treatment of changes in vegetation cover and explicit representation of crop cover are likely necessary in the CESM.

- Page 9, l34: Precise over which period the growing season length increased by 1 month. Overall the section 3.3.4 on the growing season length is not bringing much information. You could explain what contributes in the model to the change in growing season length (earlier starts or later end of the season). As for the tropic and the argument on the water use efficiency, you could provide more support by discussing how the soil moisture has evolved in the simulation with climate change.

In section 3.3.4, we will add additional details to the text to state "The NH $CO_2$ annual cycle amplitude increase resulted not only from changes in the mean temperature affecting GPP, but also from lengthening of the growing season. We found that the growing season, defined as months with negative NEP (net terrestrial carbon uptake), increased for all NH terrestrial regions by about 1 month. The overall lengthened growing seasons accounted for 1—1.3% $yr^{-1}$ of the high latitude net terrestrial carbon uptake after 2050, and up to 5% $yr^{-1}$ of the midlatitude terrestrial carbon uptake after 2100. Thus, while this is an important contributor, it is secondary to increased mid-summer GPP."

We also expand our discussion to include analysis of soil water content in CESM, per the reviewer's helpful suggestion: "The driver of the increased growing season length was different for different ecoclimate regions. For regions north of 30°N, climate change was the driver of increased growing season length. In boreal and Arctic regions, climate change extended the growing season for an additional month in the fall. In contrast, midlatitude climate change facilitated an earlier start to the growing season in the spring (Fig. 11a). $CO_2$ fertilization was the major driver of changes in the growing season length in the subtropics, while climate change had the opposite effect. This result suggests that subtropical ecosystems in CESM are near a temperature optimum, but may be water-limited. In the FullyCoupled simulation, soil water content over the top three model layers, corresponding to 0.06 m depth, decreases in the Amazon and central America by 13% on average from 1950 to 2300. In the simulation without radiative forcing (but including $CO_2$ fertilization effects), soil water content increases by 1% on average in these regions, and suggests improved water use efficiency by vegetation. Thus, increases in water use efficiency associated with increased atmospheric $CO_2$ permit longer periods of carbon uptake."

* Discussion:

- Page 10, l22-23: You mention that the CESM has probably a too strong $CO_2$ fertilization effect. This is not intuitive as you previously mentioned that CLM4CN has the lowest fertilization effect from the CMIP5 suite of model and that it provides a too low mean amplitude and mean amplitude trend for high latitude. The reasoning and conclusion should be more detailed as it is not intuitive. You can have several compensating effects so that the fertilization in the model is not too strong. Also, what is potentially missing is a discussion of the fertilization effect in CLM4CN with respect for instance to "FACE" experiment to put in perspective the results and conclusion drawn for the 23 century.

Upon further analysis, we have decided to remove this discussion from the paper as it is speculative and relies on relative, rather than absolute, trends in the amplitude.

- Page 10, l28: You mention that LUC reduced the amplitude of atmospheric $CO_2$ seasonal cycle, contrary to previous studies. You should indicate why the simulation with CESM provides new plausible information, given that you have mentioned that "treating crop as grassland" is a severe limitation (see my comment above). You have to provide some explanation on why you think the results with CESM provide a new perspective with respect to LUC. Basically what was the typical LUC that is contributing to such decrease, through which processes, . . . ?

We will clarify the discussion to state that: "In contrast, land use change in CESM reduced the atmospheric $CO_2$ mean annual cycle amplitude throughout the NH, with the largest reductions over the mid- and high latitudes. Reductions in tree cover in the FullyCoupled simulation compared to the NoLUC simulation are associated with decreases in the net carbon flux amplitude and a negative trend in the $CO_2$ annual cycle amplitude. In the FullyCoupled simulation, croplands replace the lost tree cover. Several recent papers (e.g., Zeng et al., 2014; Gray et al., 2014) suggest that agricultural amplification, facilitated by irrigation and fertilization, may be an important driver of the observed mean annual cycle trend. In the CESM, however, crop cover is currently treated as unmanaged grass and thus these agricultural practices are not explicitly modeled, and thus do not mitigate the reduction in tree cover. These results

underscore that explicit consideration of human modifications may be necessary for prognostic models both to match observations and to provide realistic predictions of future changes. We note that in CLM, development is under way to represent irrigation and fertilization in croplands in future versions of the model."

- Page 11, last paragraph: Further discussion on the fact that "the results indicate that there is no high-temperature tipping point at which terrestrial productivity declines" would be valuable. Does this mean that the temperature dependence of the maximum photosynthesis peaks at high enough temperature threshold? or is it linked to the nitrogen cycle?

We have removed this statement from the paper given that in individual pulse regions, there is a clear turnover in GPP with temperature. At high latitudes, this occurs above 300 K, and in the tropics, this occurs above 305 K. We note that temperature acclimation of GPP has recently been incorporated into CLM (Lombardozzi et al., 2015), but for CLM4, which we use, there is a clear decline. Instead, we close the paper with the "Uncertainties and future model needs" section prompted by the reviewer's comments.

"Although the results presented in this paper provide a useful look at the co-evolution of climate and the carbon cycle beyond 2100, several components of the model configuration induce substantial uncertainty into the results presented here. The lack of dynamic vegetation in this version of CESM contributes some uncertainty to these results. Tree cover is expected to expand further northward with climate change (e.g., Lloyd et al., 2005), which may contribute to the long-term increase in NEP flux amplitude within high latitude ecosystems. In contrast, drying at lower latitudes may lead to replacement of trees with grasses and subsequent decreases in NEP amplitude. Thus, the balance of these processes on the overall flux amplitude and spatial variability in the atmospheric $CO_2$ trend is uncertain. An ecosystem demography version (CLM-ED) is currently being developed that would permit successional patterns in response to environmental change. We consider the documentation of trends in the static-vegetation configuration presented in this manuscript to be a crucial first step toward eventually determining the sensitivity of land-atmosphere biogeochemical coupling in more sophisticated, future configurations of the CESM model.

The lack of permafrost dynamics likely has a large impact on $CO_2$ annual cycle trends, especially later in the simulation when global mean temperature has increased by over 10 K in the fully coupled simulation. Ongoing model development in CESM includes improved representation of permafrost carbon (Koven et al., 2015), and thus future model configurations will provide an improved tool for investigating a process that may provide one of the tipping points we identify in our key science questions."

**Additional References**

Ito, A., Huntzinger, D. N., Schwalm, C., Michalak, A. M., Cook, R., King, A. W., Mao, J., Wei, Y., Mac Post, W., and Wang W.: Decadal trends in the seasonal-cycle amplitude of terrestrial $CO_2$ exchange resulting from the ensemble of terrestrial biosphere models, Tellus B, 68, 2016.

Lombardozzi, D. L., Bonan, G. B., Smith, N. G., Dukes, J. S., and Fisher, R. A.: Temperature acclimation of photosynthesis and respiration: A key uncertainty in the carbon cycle-climate feedback, Geophys. Res. Lett., 42, 8624–8631, doi:10.1002/2015GL065934, 2015.

Long, M. C., Lindsay, K., Peacock, S., Moore, J. K., and Doney, S. C.: Twentieth-Century Oceanic Carbon Uptake and Storage in CESM1(BGC), J. Climate, 26, 6775–6800, doi:10.1175/JCLI-D-12-00184.1, 2013.

Lloyd, A. H.: Ecological histories from Alaskan tree lines provide insight into future change, Ecology, 86 1687–1695, doi:10.1890/03-0786, 2005.

Zhao, F., Zeng, N., Akihiko, I., Asrar, G., Friedlingstein, P., Jain, A., Kalnay, E., Kato, E., Koven, C. D., Poulter, B., Rafique, R., Sitch, S., Shu, S., Stocker, B., Viovy, N., Wiltshire, A., and Zaehle, S.: Role of $CO_2$, climate and land use in regulating the seasonal amplitude increase of carbon fluxes in terrestrial ecosystems: a multimodel analysis, Biogeosciences Discuss., doi:10.5194/bg-2016-121, in review, 2016.

---

## Author Response (AR2)

February 10, 2017

Dear Laurent,

Thank you for accepting our manuscript following minor revisions. We have edited the manuscript per your and the reviewer's suggestions, and have corrected the figure captions as necessary. The color bars in Figure 8 (Figure 9 in the revised manuscript), and in the new Figure 15, do not have distinct divisions as in Figures 6—8, because all 351 years are shaded separately. Therefore, the color scheme is depicted as a gradient, with the ticks marking the approximate location 1950, 2000, …, 2300.

Sincerely,

Jessica Liptak

We thank the reviewer for her or his additional insight on our manuscript. Here, we respond point-by-point to the reviewer's comments. The reviewer comments are in Times New Roman, and our responses are in Arial font.

I thank the authors for their detailed responses to my comments. They made a substantial effort to account for most of the concerns I raised with thorough responses and the needed clarifications in the text for a better understanding of the paper objectives and methodology (in particular the pulse response approach).

The description of the context of the study has been significantly improved (especially with respect to recent publications) with a refinement of the focus of the paper; the objectives are now less on the understanding of the drivers of the atmospheric $CO_2$ amplitude increase but more on how this observational constraint could be used with a couple carbon - climate model and how the driver of the $CO_2$ amplitude increase may change in the 2300 time horizon.

Although I still believe that current model structure (related to CESM1 version that is used) limits the biogeochemical conclusions that could be drawn for the horizon 2100 - 2300 (lack of permafrost, dynamic vegetation and ecosystem management (crops)), the paper can bring some interesting methodological/theoretical contributions to the existing literature around the valorisation of the atmospheric $CO_2$ amplitude changes. With this context in mind I would encourage the author to slightly improve the text to emphasize especially in the discussion on the methodological aspects and to draw further recommendations for other ESM modeling groups with respect to how to use the constraint from atmospheric $CO_2$ amplitude growth.

We have modified Section 4.1 to discuss the changes in the drivers of the $CO_2$ annual cycle amplitude through 2300 in more detail, and include suggestions for model evaluation based on our results.

Note also that the discussion contains now only one sub-section « 4.1 » which seems strange.

Section 4 is now split subsections **4.1 CO$_2$ annual cycle amplitude trends and applications for model evaluation**, and **4.2 Uncertainties and future model needs**

Finally as a little remark it seems to me that the discussion on the Water Use Efficiency, although welcome, should be more precise. The authors conclude that from a simulation without radiative forcing (but including CO2 fertilization effects), soil water content increases by 1% on average and suggest improved WUE by vegetation. This seems not so obvious as changes in precipitation, relative humidity, … complicate the relation and they probably have the model output to diagnose the effective WUE or even the intrinsic one.

We have added an analysis of changes in NH subtropical water use efficiency (WUE) to section 3.3.4. using Central America as an example.

Manuscript changes for the revised version of *Drivers of Multicentury Trends in the Atmospheric CO$_2$ Mean Annual Cycle in a Prognostic ESM* (J.Liptak, G. Keppel-Aleks, K. Lindsay)

The marked-up document shows changes between the previous manuscript version and revised manuscript, with addition indicated by blue underlined text, and deletion indicated by red strikethrough text.

Major changes are listed below:

1. 1950—1959 is now described as the "baseline" rather than "present-day" reference period
2. We diagnosed water use efficiency (WUE) and modified the discussion of water limitation in Section 3.3.4
3. The previous Figure 15 showing soil water changes in the Amazon and Central America has been removed and replaced it with a figure depicting relationships among NEP, WUE, and GPP in Central America
4. Section 4 Discussion and Conclusions contains two subsections: *4.1 CO$_2$ annual cycle amplitude trends and applications for model evaluation*, and *4.2 Uncertainties and future model needs*
5. Section 4.1 more clearly describes the utility of the results in model evaluation
6. Table 2 has been modified to include the ecoclimate regions shown in Figure 1, and the region numbers have been removed
7. Figure 13 now depicts decadal averages ending in 2000, 2050, 2100, 2150, 2200, 2250, and 2300 to align with the time periods referenced in the text

[revised manuscript text omitted]